# Cooperative and acute inhibition by multiple C-terminal motifs of L-type Ca$^{2+}$ channels

Nan Liu[1†], Yaxiong Yang[1†], Lin Ge[1†], Min Liu[1], Henry M Colecraft[2], Xiaodong Liu[1,3,4]*

[1]X-Lab for Transmembrane Signaling Research, Department of Biomedical Engineering, School of Medicine, Tsinghua University, Beijing, China; [2]Department of Physiology and Cellular Biophysics, Columbia University, New York, United States; [3]School of Life Sciences, Tsinghua University, Beijing, China; [4]IDG/McGovern Institute for Brain Research, Tsinghua University, Beijing, China

**Abstract** Inhibitions and antagonists of L-type Ca$^{2+}$ channels are important to both research and therapeutics. Here, we report C-terminus mediated inhibition (CMI) for Ca$_V$1.3 that multiple motifs coordinate to tune down Ca$^{2+}$ current and Ca$^{2+}$ influx toward the lower limits determined by end-stage CDI (Ca$^{2+}$-dependent inactivation). Among IQ$_V$ (preIQ$_3$-IQ domain), PCRD and DCRD (proximal or distal C-terminal regulatory domain), spatial closeness of any two modules, *e.g.*, by constitutive fusion, facilitates the trio to form the complex, compete against calmodulin, and alter the gating. Acute CMI by rapamycin-inducible heterodimerization helps reconcile the concurrent activation/inactivation attenuations to ensure Ca$^{2+}$ influx is reduced, in that Ca$^{2+}$ current activated by depolarization is potently (~65%) inhibited at the peak (full activation), but not later on (end-stage inactivation, ~300 ms). Meanwhile, CMI provides a new paradigm to develop Ca$_V$1 inhibitors, the therapeutic potential of which is implied by computational modeling of Ca$_V$1.3 dysregulations related to Parkinson's disease.

*For correspondence:
liuxiaodong@tsinghua.edu.cn

†These authors contributed equally to this work

**Competing interests:** The authors declare that no competing interests exist.

## Introduction

L-type Ca$^{2+}$ channels (LTCC or Ca$_V$1 channels) play pivotal roles in numerous physiological functions by mediating Ca$^{2+}$ influx and membrane excitability (*Striessnig et al., 2014*). Among four isoforms of Ca$_V$1.1–Ca$_V$1.4 in LTCC family, Ca$_V$1.3 channels exhibit unique biophysical properties (*Lieb et al., 2014*; *Xu and Lipscombe, 2001*) involving diverse regulatory mechanisms (*Ben-Johny and Yue, 2014*; *Christel and Lee, 2012*; *Huang et al., 2012*; *Tan et al., 2011*). Widely distributed in various excitable tissues, including both cardiovascular and nervous systems (*Brandt et al., 2003*; *Mangoni et al., 2003*; *Nemzou et al., 2006*; *Striessnig, 2007*), Ca$_V$1.3 is also involved in a wide spectrum of pathology, including multiple inheritable diseases (*Pinggera and Striessnig, 2016*; *Platzer et al., 2000*). As one prominent exemplar of its pathophysiological linkage, Ca$_V$1.3 expressed in substantia nigra pars compacta (SNc) neurons tightly controls the autonomous sub-threshold Ca$^{2+}$ oscillations which have been considered to underlie the pathophysiology of Parkinson's disease (PD) (*Chan et al., 2007*; *Guzman et al., 2009*, *2010*).

Ca$_V$1.3 is tuned by its own distal carboxyl tail (DCT) to compete with apoCaM (Ca$^{2+}$-free calmodulin), which is pre-associated with the carboxyl terminus of the channel at preIQ$_3$-IQ domain (denoted as IQ$_V$). Closely involved in both apoCaM and Ca$^{2+}$/CaM binding (*Jurado et al., 1999*), IQ$_V$ plays important roles in channel functions (*Ben Johny et al., 2013*; *Liu et al., 2010*; *Singh et al., 2006*; *Wahl-Schott et al., 2006*). The competitive tuning is highly regulated by fluctuations of

**eLife digest** All cells need calcium ions to stay healthy, but having too many calcium ions can interfere with important processes in the cell and cause severe problems. Proteins known as calcium channels on the cell surface allow calcium ions to flow into the cell from the surrounding environment. Cells carefully control the opening and closing of these channels to prevent too many calcium ions entering the cell at once. $Ca_V1.3$ channels are a type of calcium channel that are important for the heart and brain to work properly. Defects in $Ca_V1.3$ channels can lead to irregular heart rhythms and neurodegenerative diseases such as Parkinson's disease.

Studies have shown that part of the $Ca_V1.3$ channel that sits inside the cell – known as the "tail" – responds to increases in the levels of calcium ions inside the cell by closing the channel. The tail region of $Ca_V1.3$ contains three modules, but how these modules work together to regulate channel activity is not clear.

Liu, Yang et al. investigated whether the three modules need to be physically connected to each other in the channel protein. For the experiments, several versions of the protein were constructed with different combinations of tail modules being directly linked as part of the same molecule or present as separate molecules. When any two modules were directly linked, the third module could bind to them and this was enough to close the $Ca_V1.3$ channel. However, the channel did not close if the modules were totally isolated from each other as three separate molecules.

Certain types of neurons in the brain produce electrical signals in a rhythmic fashion that depends on $Ca_V1.3$ channels. In Parkinson's disease, increased movement of calcium ions into these neurons via $Ca_V1.3$ channels interferes with the rhythms of the signals and can cause these cells to die. Liu, Yang et al. performed computer simulations to analyse the effects of closing $Ca_V1.3$ channels in these neurons. The results suggest that this can restore normal rhythms of electrical activity and prevent these cells from dying.

The next step is to understand the molecular details of how the tail region closes $Ca_V1.3$ channels and its role in healthy and diseased cells. This may lead to new ways to block $Ca_V1.3$ channels in different types of diseases.

[apoCaM], the strength of particular DCT isoforms, or the apoCaM affinity with $IQ_V$ (*Adams et al., 2014*; *Bazzazi et al., 2013*; *Liu et al., 2010*). DCT consists of two putative α-helical domains—a proximal and a distal C-terminal regulatory domain (termed PCRD and DCRD, respectively) (*Hulme et al., 2006*). Positively charged PCRD could coordinate with negatively charged DCRD, in the context of $Ca_V1.3$ (*Liu et al., 2010*), to regulate effective CaM affinity to the $IQ_V$ region of the channel. Besides the reported inhibition on $Ca^{2+}$-dependent inactivation (CDI) (*Liu et al., 2010*; *Singh et al., 2006*; *Wahl-Schott et al., 2006*), several studies suggest that DCT also concurrently attenuates voltage-gated activation (VGA), reducing the maximum open probability and positively shifting the voltage dependence (*Hulme et al., 2006*; *Lieb et al., 2014*; *Liu et al., 2016*; *Scharinger et al., 2015*; *Singh et al., 2008*; *Wahl-Schott et al., 2006*). Altogether, this competitive tuning of $Ca_V1$ gating (activation/inactivation) emerges as a new modality distinct from conventional inhibitions such as intensively-studied $Ca^{2+}$ channel blockers (CCBs) that normally only reduce VGA (*Hockerman et al., 1997*), or $Ca^{2+}$/CaM-triggered conformational changes that induce CDI (*Ben-Johny and Yue, 2014*). However, several key matters still remain to be clarified before such new CMI (C-terminus Mediated Inhibition) could be fully established. First, PCRD, DCRD and $IQ_V$ seemingly work cooperatively to mediate CMI as suggested by aforementioned analyses, but the exact interrelationships among these three motifs are yet to be elucidated. Second, CMI is supposed to act on channels in an acute manner similar to widely-applied CCBs or like interventions, but direct evidence is still lacking. Third, CMI is expected to reduce the overall $Ca^{2+}$ influx; however, it is still unclear whether and how CMI is able to ensure the actual inhibition when both VGA and CDI are attenuated, apparently leading to contradictory effects on $Ca^{2+}$ influx.

Embarking on these frontiers regarding CMI, we unveiled the principle of cooperation among the three key C-terminal motifs, with which we developed chemical-inducible CMI for $Ca_V1.3$ channels. We then resolved all the aforementioned questions crucial to establish CMI, and gained new insights

into $Ca_V1.3$ gating by comparing CMI and CDI. In addition, with computational modeling of $Ca^{2+}$ oscillations and pacemaking behaviors under the control of $Ca_V1.3$ channels in SNc neurons, we explored the potentials of CMI-based inhibitors as therapeutic interventions for $Ca_V1.3$-related PD.

## Results

### Provisional CMI concurrently attenuates both CDI and VGA

Alternative splicing at the carboxyl terminus of $\alpha_{1D}$ results into two important native variants of long and short forms: $\alpha_{1DL}$ (exon 42) and $\alpha_{1DS}$ (exon 42A). Short variant of $\alpha_{1DS}$ terminates shortly after the $IQ_V$ motif and lacks almost entire DCT domain, and its $Ca^{2+}$ current ($I_{Ca}$) exhibited strong CDI and VGA (*Figure 1A*). Regarding CDI, $\alpha_{1DS}$ still contains all the core elements including the $IQ_V$ domain for apoCaM pre-association and the EF-hand motifs and N-terminal $Ca^{2+}$/CaM binding NSCaTE motif (*Dick et al., 2008*) for CDI transduction. $I_{Ca}$ upon depolarization rapidly decayed (second row, red trace), indicative of strong CDI. In contrast, $Ba^{2+}$ current ($I_{Ba}$) decayed rather slowly (trace in grey) representing the background voltage-dependent inactivation (VDI). Thus, the fraction of peak current ($I_{peak}$) remaining after depolarization for 50 ms (third row, $r_{50}$) is closely correlated with inactivation, with the difference between the $r_{50}$ profiles in $Ba^{2+}$ and $Ca^{2+}$ serving as the ideal index of CDI. In practice, due to rather weak VDI of $Ca_V1.3$ at 50 ms ($r_{50,Ba}$ close to 1) (*Tadross et al., 2010*), thus for simplicity the value of $1-r_{50,Ca}$ at $-10$ mV was taken as the index of CDI strength ($S_{Ca}$). VGA was evaluated by peak current density $J_{peak}$ ($pA/pF$), measured at different voltages while excluding potential artifacts due to cell-size factor (capacitance). Throughout this study, $S_{Ca}$ and $J_{Ca}$ ($J_{peak}$ at $-10$ mV) served as the quantitative indices for CDI and VGA respectively. Thus, by way of DCT competition against apoCaM, the provisional CMI was expected to attenuate both $S_{Ca}$ and $J_{Ca}$, as suggested by prior studies (*Liu et al., 2010*; *Singh et al., 2008*, *2006*; *Tan et al., 2011*, *2012*; *Wahl-Schott et al., 2006*). However, most of these reports focused on either CDI or VGA only; or even when both were studied for whole-cell $I_{Ca}$, CDI and VGA were separately evaluated with different experimental groups. Here, we conducted a side-by-side analysis for each variant or condition, not only more confirmative since concurrent changes in CDI and VGA would be mutually supportive, but also critical to later insights into CMI mechanisms and significance originated from such concurrency. Technically, compared to VGA and $J_{Ca}$, CDI and $S_{Ca}$ are more robust so more favorable since CDI of $Ca_V1.3$ is independent of global $Ca^{2+}$ so insensitive to $Ca^{2+}$-buffer capacities and variations in channel expressions partly due to transient transfections (*Dick et al., 2008*; *Tadross et al., 2008*). In general, for data and analyses we provided for both CDI and VGA, CDI ($S_{Ca}$) was considered as the major index for CMI evaluations and VGA as the supplement.

To establish CMI, several key issues as introduced earlier regarding its mechanisms of action and potential applications need to be clarified. We decided to employ homologous $DCT_F$ from $Ca_V1.4$ ($\alpha_{1F}$) with the strongest DCT tuning effects (stronger than $DCT_D$) among the $Ca_V1$ members (*Figure 1—figure supplement 1*), to construct the channel variant by fusing $PCRD_F$-$DCRD_F$ onto $\alpha_{1DS}$ right after $IQ_V$ motif; meanwhile, since the potency of DCT effects is mainly determined by DCRD (*Liu et al., 2010*), PCRD from either $Ca_V1.3$ or $Ca_V1.4$ supposedly makes no or little difference and thus we did not distinguish these two PCRD isoforms in this work. Such channel variant was simply denoted as $\alpha_{1DS}$-PCRD-DCRD, which represents CMI intrinsic to the long variant of $Ca_V1.3$ ($\alpha_{1DL}$) but with higher potency since strong $DCRD_F$ replaces endogenous $DCRD_D$. Compared with $\alpha_{1DS}$ control, $S_{Ca}$ and $J_{Ca}$ from $\alpha_{1DS}$-PCRD-DCRD were substantially attenuated, which provided ample dynamic ranges (highlighted by green areas) to explore CMI effects on CDI/VGA and its mechanisms (*Figure 1B*).

### Differential roles of DCRD and PCRD unveiled by low [apoCaM]

With such baseline CMI effects of PCRD-DCRD, we went further to examine whether both PCRD and DCRD are required, by fusing only PCRD or DCRD onto $\alpha_{1DS}$. Since DCRD might need some structural freedom as in DCT or PCRD-DCRD, a glycine linker of different length ($G_X$) was inserted between DCRD and $IQ_V$ to construct $\alpha_{1DS}$-$G_X$-DCRD, mimicking the configurations that permit potent effects. From all these channel variants, we concluded no alteration in CDI in comparison with the $\alpha_{1DS}$ control, similarly from VGA results and analyses (*Figure 2A* and *Figure 2—figure*

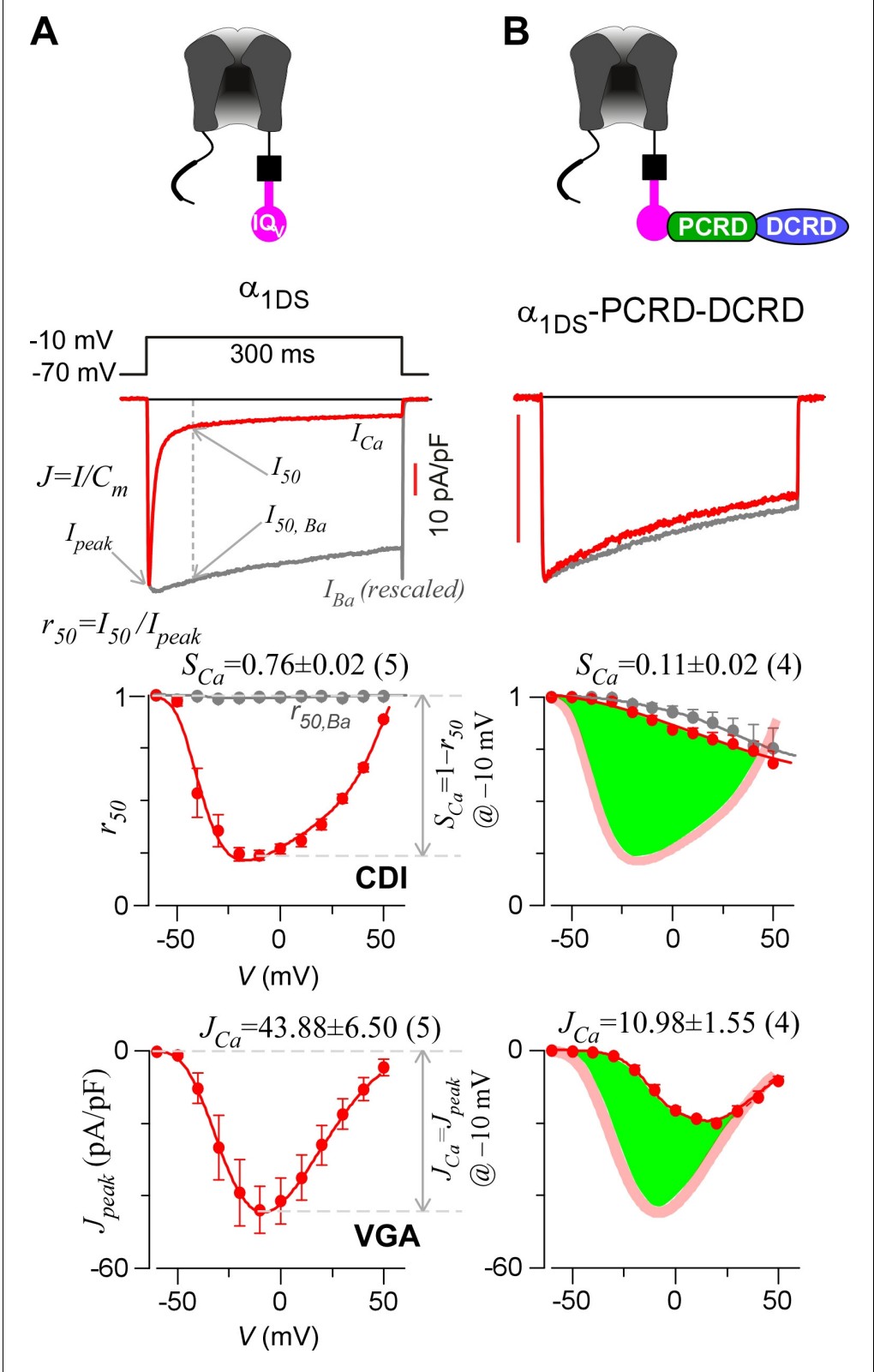

**Figure 1.** Inhibition of $Ca_V1.3$ gating by carboxyl terminal motifs. (**A**) Parameters and indices illustrated by the control group of short $Ca_V1.3$ channels. Representative current exemplars ($Ca^{2+}$ current $I_{Ca}$ in red, with the scale bar in red; $Ba^{2+}$ current $I_{Ba}$ in gray, rescaled) were shown for $\alpha_{1DS}$ at the membrane potential (*V*) of −10 mV, with the amplitudes measured at the time of peak ($I_{peak}$) and 50 ms ($I_{50}$). Inactivation profiles across the full range of *V*

*Figure 1 continued*

for $I_{Ba}$ and $I_{Ca}$ were quantified by the remaining current at 50 ms ($r_{50}$), in ratio between $I_{50}$ and $I_{peak}$. The CDI strength was quantified by $1-r_{50,Ca}$ or $S_{Ca}$, serving as one of the major indices. Based on $Ca^{2+}$ current density normlized to the cell capacitance ($C_m$), VGA in $Ca^{2+}$ was profiled by $J_{peak}$ (in pA/pF), with the $J_{peak}$ amplitude at $-10$ mV or $J_{Ca}$ as the other major index. (**B**) In contrast to $\alpha_{1DS}$ channels with pronounced CDI ($S_{Ca}$) and strong VGA ($J_{Ca}$), $\alpha_{1DS}$-PCRD-DCRD incorporating all the three motifs of $IQ_V$, PCRD and DCRD exhibited strong inhibitions on both CDI and VGA (less pronounced U-shape or V-shape), indexed with $S_{Ca}$ and $J_{Ca}$ (smaller values) respectively. Thick semi-transparent lines in red depict the CDI and VGA profiles of the $\alpha_{1DS}$ control group (**A**); and the differences in profiles (green areas as visual cues) illustrate the potency of CMI.
The following figure supplement is available for figure 1:

**Figure supplement 1.** Sequence alignment of carboxyl termini for L-type $Ca_V1$ channels.

*supplement 1A*). In general, current densities should be more carefully scrutinized before any conclusive claim on VGA; and effects on $J_{peak}$ profiles and $J_{Ca}$ values normally need to be backed up by evaluations on concurrent CDI. For instance, current densities of $\alpha_{1DS}$-$G_X$-DCRD (*Figure 2—figure supplement 1A*, the three columns on right) seemed different from the control, potentially interpreted as mild CMI; however, considering that the changes in CDI of these channels were rather small, the mild changes in VGA should be considered as variations instead of conclusive attenuations. Statistical test of significance (Student's t-test) was performed subsequently to help identify significant CMI effects. Based on the fact that all the configurations under test failed to mediate $S_{Ca}$ and $J_{Ca}$ attenuations, all the three motifs of PCRD, DCRD and $IQ_V$ might be required for effective CMI, consistent with the preliminary analyses on DCT subsegments from $Ca_V1.3$ and $Ca_V1.4$, in which the segment between PCRD and DCRD ('B' region) was dispensable but neither PCRD nor DCRD could be excluded (*Liu et al., 2010*).

Notably, these channel variants expressed in HEK cells were quantified under normal concentrations of $Ca^{2+}$-free CaM ([apoCaM]), in which the three-motif complex of $IQ_V$/PCRD/DCRD as the core of CMI was perturbed by apoCaM that tends to pre-associate with $IQ_V$ of the channel. To alleviate the interference with the potential cooperation in forming the threesome complex, we managed to substantially reduce [apoCaM] in the cells by overexpressing $BSCaM_{IQ}$, a strong apoCaM buffer (*Black et al., 2006*), to help identify the minimum requirement for CMI (*Figure 2B*). In contrast to strong CDI in normal [apoCaM] evidenced from different channel variants (*Figure 2A*), with [apoCaM] being substantially reduced (but not lower than a realistic limit), surprisingly, $\alpha_{1DS}$-$G_{24}$-DCRD and all the other DCRD-containing variants exhibited much attenuated CDI indicative of effective CMI (*Figure 2C,D* and *Figure 2—figure supplement 1*). Meanwhile, the stereotyped ultrastrong CDI normally observed from $\alpha_{1DS}$ or $\alpha_{1DS}$-PCRD was nearly unaltered by apoCaM buffers, in that under the residue [apoCaM] these channels were still able to be pre-bound with apoCaM considering the effective affinity between apoCaM and the channel ($IQ_V$ or $IQ_V$-PCRD) was reasonably high when there was no interference. Hence, mechanistically only DCRD and $IQ_V$ are required by CMI so that PCRD becomes dispensable if [apoCaM] is low, unveiling the differential roles of PCRD and DCRD in the potential cooperation for CMI induction.

## Cooperative scheme of CMI consists of three major combinations

The data thus far suggest that CMI mainly relies on DCRD/$IQ_V$ binding to overcome free [apoCaM] of substantial level in normal cell conditions; and PCRD could facilitate CMI potentially by enhancing the binding of DCRD/$IQ_V$ and thus the competition against apoCaM. However, the trio failed to attenuate CDI when they were separately expressed as three individual peptides (*Figure 3A–C*), even when PCRD, DCRD and $IQ_V$ (contained in $\alpha_{1DS}$) were all well expressed in the same cell as confirmed with fluorescent tags (*Figure 3—figure supplement 1*). By examining VGA ($J_{Ca}$) with aforementioned precautions, we also concluded that no CMI effect on $\alpha_{1DS}$ was detectable with all the three groups of peptides: PCRD and DCRD, PCRD only, and DCRD only. When [apoCaM] was reduced with chelator $BSCaM_{IQ}$ as in *Figure 2*, CDI remained ultrastrong and indistinguishable from $\alpha_{1DS}$ control for all the three test groups (*Figure 3D–F*), and VGA analyses came to the same conclusions consistently (*Figure 3—figure supplement 2D–F*).

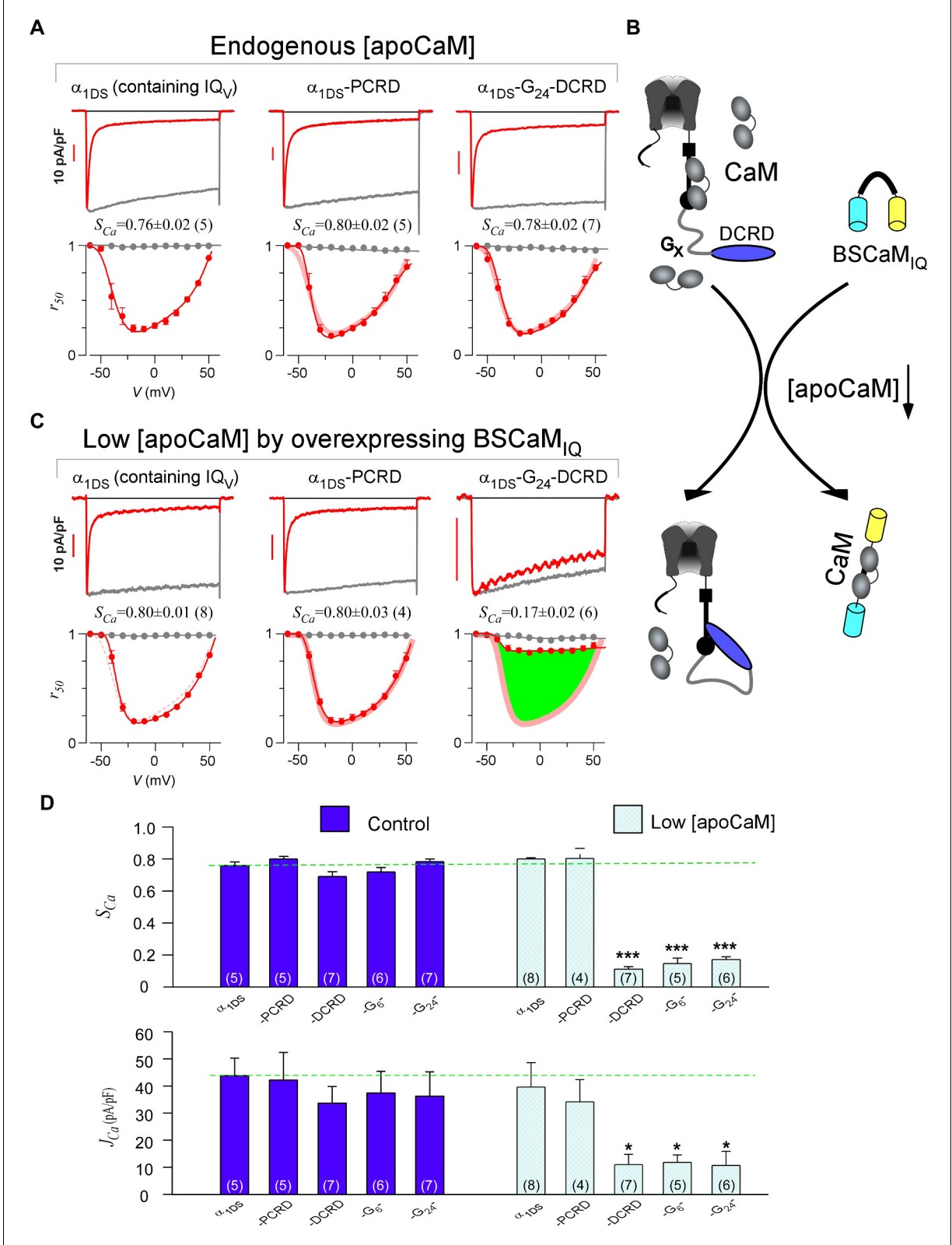

**Figure 2.** DCRD is indispensable and sufficient to induce CMI in low [apoCaM]. (**A**) All the channel variants including $\alpha_{1DS}$, $\alpha_{1DS}$-PCRD and $\alpha_{1DS}$-G$_{24}$-DCRD exhibited strong CDI, indexed with $S_{Ca}$. (**B**) Schematic illustration for the strategy to explore the minimum requirement of CMI. For $\alpha_{1DS}$-G$_X$-DCRD containing glycine linkers (G$_X$) of different length (0, 6 or 24), BSCaM$_{IQ}$ that binds apoCaM was overexpressed to downregulate endogenous [apoCaM], in hope to promote the binding of DCRD with the channel (IQ$_V$). (**C**) With low [apoCaM] by overexpressing BSCaM$_{IQ}$, $\alpha_{1DS}$-G$_{24}$-DCRD

*Figure 2 continued on next page*

Figure 2 continued

channels exhibited much attenuated CDI, evidenced by $I_{Ca}$ trace, $S_{Ca}$ value and $r_{50}$ profile (green area indicating the potency), in contrast to ultrastrong CDI of $\alpha_{1DS}$ or $\alpha_{1DS}$-PCRD, both lacking DCRD. (D) Statistical summary of CDI ($S_{Ca}$) and VGA ($J_{Ca}$) in endogenous (control) or low [apoCaM] for all channel variants of $\alpha_{1DS}$, $\alpha_{1DS}$-PCRD and $\alpha_{1DS}$-$G_X$-DCRD, with additional information in *Figure 2—figure supplement 1*. Notably, for the three $\alpha_{1DS}$-$G_X$-DCRD variants, both CDI and VGA were concurrently and significantly attenuated. Statistical significance was evaluated and indicated (p<0.001, ***; p<0.05, *).

The following figure supplement is available for figure 2:

**Figure supplement 1.** Full CDI and VGA profiles of different channel variants in endogenous and low [apoCaM].

Up to here, we established two extreme cases for CMI, *i.e.*, either all the three motifs were fused together (strong CMI), or the three were totally isolated to each other (no CMI). To explore intermediate conditions between the two extremes, we physically linked every two modules out of the three to coexpress with the third one, yielding three possible combinations in total. And they were, PCRD-DCRD peptides to coexpress with $IQ_V$ (contained in $\alpha_{1DS}$) (*Figure 4A*, combination I), channels containing $IQ_V$-PCRD to coexpress with DCRD (*Figure 4B*, combination II), and channels containing linked $IQ_V$ and DCRD to coexpress with PCRD (*Figure 4C*, combination III). Interestingly, all the three combinations exhibited CMI, strongly attenuating both CDI and VGA as illustrated by green areas indicating the altered $r_{50}$ and $J_{peak}$ profiles compared to $\alpha_{1DS}$ control (*Figure 4A—C*). Combination I also provided a great chance to revisit the assumption we made earlier regarding interchangeable $PCRD_D$ and $PCRD_F$ from $Ca_V1.3$ and $Ca_V1.4$, respectively. Instead of dealing with two different channel variants, here the same $\alpha_{1DS}$ served as the control to reliably evaluate $PCRD_D$-DCRD *vs* $PCRD_F$-DCRD peptides. And our assumption was validated by the negligible differences in VGA and CDI between the two PCRD-DCRD peptides (*Figure 4—figure supplement 1*). In previous reports, combinations of I and II were already suggested when studying endogenous DCT in $Ca_V1.3$ ($\alpha_{1DL}$) and $Ca_V1.4$ (*Singh et al., 2008, 2006*). In addition, PCRD (combination III) was able to induce pronounced CMI effects as we first observed in this work, which also completed all the three possible combinations of peptide-mediated CMI.

Live-cell FRET two-hybrid assays demonstrated that DCRD bound $IQ_V$-PCRD with higher affinity ($K_d = 1135$, units in donor-cube fluorescence intensity) than that between DCRD and $IQ_V$ ($K_d = 4700$), regardless of whether PCRD as the standalone peptide was also present ($K_d = 4624$) (*Figure 4D*). Owing to the high affinity between DCRD and $IQ_V$-PCRD, DCRD overexpression nearly abolished the binding between apoCaM and $IQ_V$-PCRD (*i.e.*, apoCaM/channel pre-association) as demonstrated in the plot of $FR$-$D_{free}$ (FRET ratio *vs* free donor concentrations) (*Figure 4—figure supplement 1A*); meanwhile, without the presence of PCRD, apoCaM was still able to bind $IQ_V$, but the DCRD perturbation was much less effective (*Figure 4—figure supplement 1B*). The electrostatic interaction between PCRD and DCRD hinted by $Ca_V1.2$ (*Hulme et al., 2006*) seemed generalizable but was not detected with the FRET pairs based on $Ca_V1.4$ DCT (*Liu et al., 2010*). Both this putative PCRD/DCRD interaction and the binding between $IQ_V$ and DCRD should be weaker and secondary compared to the strong binding between $IQ_V$-PCRD and DCRD (*Figure 4D*). This explained why the potential contributions of these secondary interactions may or may not be evidenced, depending on whether the trio complex would come into being in particular settings. Our FRET binding analyses are consistent with the functional data shown earlier, thus providing additional information for the collaborations among CMI motifs. For instance, coexpression of all the three but separate components would not produce CMI (*Figure 3A*). Also, DCRD strongly attenuated $\alpha_{1DS}$-PCRD but not $\alpha_{1DS}$ control (*Figure 4B vs Figure 3C*).

Taken together, a cooperative scheme of CMI under physiological [apoCaM] in the cell was unveiled by functional and binding analyses (*Figure 4E*), which depicted that if any two components have enough spatial closeness or intimacy, *e.g.*, by prearranged linkage, it would be sufficient for the third component to form the 'trio' complex and thus induce effective CMI (combinations of I, II and III, highlighted in grey ellipses). Also, the fusion of all the three components together mimics the intrinsic CMI in $\alpha_{1DL}$ (positive control, *Figure 1B*); and expression of three components separate to each other represents the negative control with no CMI (*Figure 3A*).

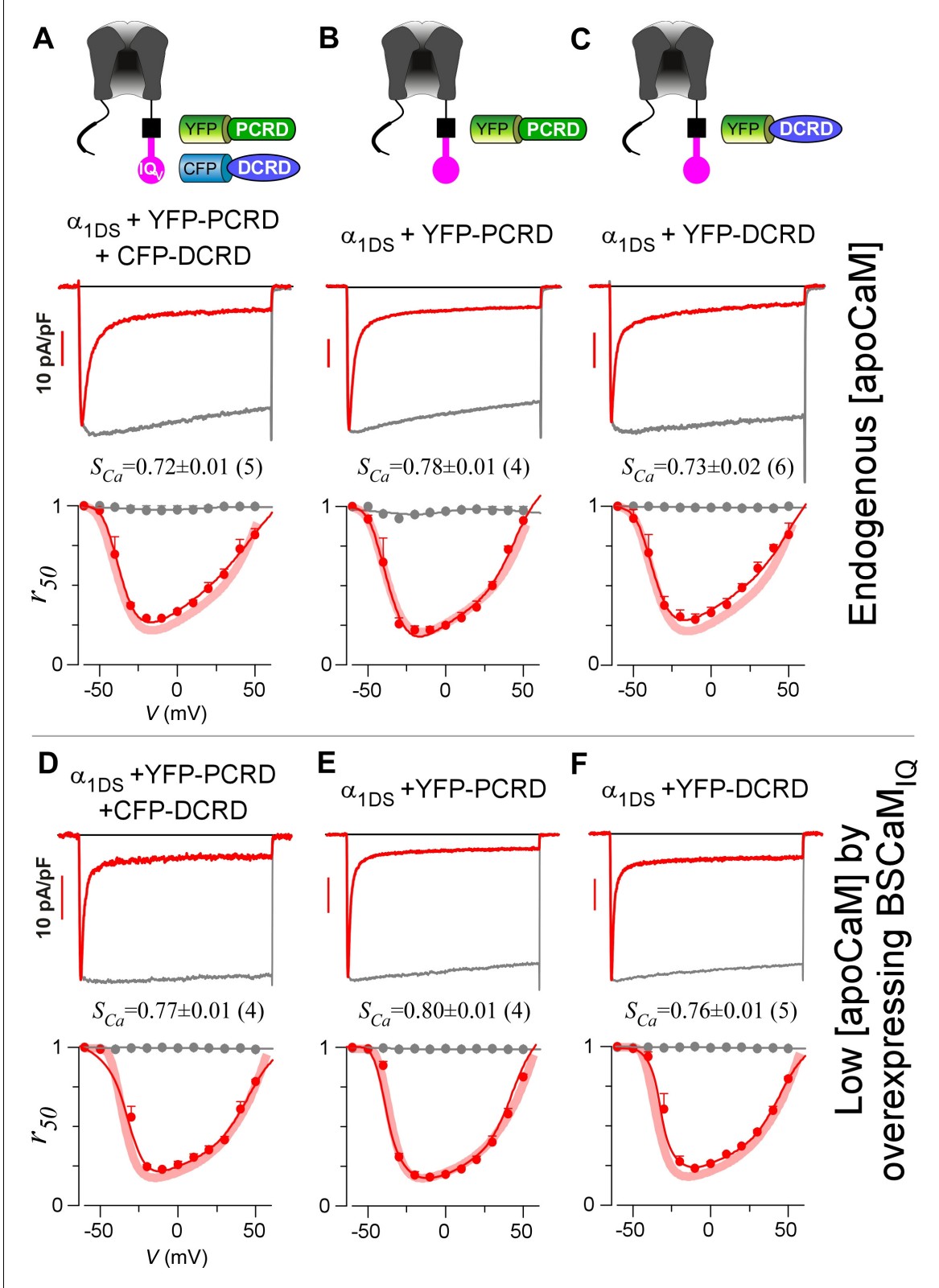

**Figure 3.** Individual DCRD peptides expressed separately from $IQ_V$ are unable to induce CMI. (**A**) PCRD and DCRD tagged with fluorescent proteins were coexpressed with $\alpha_{1DS}$ as separate peptides. The presence of both YFP-PCRD and CFP-DCRD peptides in the same cell was confirmed under a fluorescence microscope (***Figure 3—figure supplement 1***). Under normal [apoCaM] in cells, no CMI effect was observed from $I_{Ca}$ trace exhibiting CDI similarly as $\alpha_{1DS}$ control, confirmed by comparable $S_{Ca}$ values and indistinguishable $r_{50}$ profiles. (**B** and **C**) Experiments and analyses were
*Figure 3 continued on next page*

*Figure 3 continued*

performed in normal [apoCaM] similarly as (A), except that only YFP-PCRD (B) or only YFP-DCRD (C) was expressed with $\alpha_{1DS}$ channels. Both resulted into strong CDI, with $S_{Ca}$ and $r_{50}$ indistinguishable from $\alpha_{1DS}$ control. (D−F) When free [apoCaM] was substantially reduced by overexpressing $BSCaM_{IQ}$ (apoCaM buffers), the above three cases in (A−C) were re-examined. Exemplar $I_{Ca}$ traces exhibited strong CDI as quantified by $S_{Ca}$ values and $r_{50}$ profiles, similarly as the control group of $\alpha_{1DS}$ in low [apoCaM] (semitransparent line in red).

The following figure supplements are available for figure 3:

**Figure supplement 1.** To confirm the presence of both PCRD and DCRD peptides in single cells.
**Figure supplement 2.** PCRD and DCRD as separate peptides are unable to inhibit VGA.

## Acute CMI based on rapamycin-inducible heterodimerization and cooperation

Based on such cooperation-dependent CMI (*Figure 4E*) and rapamycin-mediated heterodimerization (*Yang et al., 2013*, *2007*), we devised the strategy to implement drug-inducible CMI, illustrated in the design of version 1 (*Figure 5A*). FKBP (rapamycin-binding protein) and FRB fragment (FKBP and rapamycin binding domain of the kinase mTOR) as the tags were fused onto DCRD and PCRD, respectively. A small immunosuppressant molecule rapamycin that binds both FRB and FKBP was applied to link one FKBP-DCRD together with one PCRD-FRB, aiming to bind to the $IQ_V$ domain of $\alpha_{1DS}$ to compete off apoCaM and thus induce CMI according to the scheme of combination I (*Figure 4E*). This design, if successful, should be directly applicable to native $Ca_V1.3$ since no modification is needed at the channel side. Attracted by such potentials, we implemented and validated this rapamycin-inducible peptide-mediated CMI in HEK cells with $\alpha_{1DS}$. In contrast to the stable time-course profiles of the control group (*Figure 5B*), CDI ($S_{Ca}$) and VGA ($I_{peak}$) were both rapidly attenuated upon applying 1 μM rapamycin. Within tens of seconds, dimerization between FKBP-DCRD with PCRD-FRB started to attenuate $I_{Ca}$ as evidenced by time-dependent decays (*Figure 5C*).

Encouraged by the first prototype of inducible CMI, we then proceeded to enhance the potency from moderate (as in version 1) to ultrastrong attenuation similarly as in *Figure 1B*. We reasoned that the underperformance here should be mainly due to the low effective concentration of the linked peptides if closely comparing constitutive (*Figure 4A*) and the rapamycin-inducible CMI (*Figure 5C*). The upgrade version (version 2) was designed to enhance concentration (local to channels) and thus the potency of CMI peptides. Membrane-targeting motif Ras (*Yang et al., 2007*) was used to construct FRB-CFP-Ras, which would bind DCRD/PCRD tagged with FKBP through rapamycin-inducible heterodimerization. As demonstrated by confocal fluorescence imaging, in about two minutes YFP-FKBP-DCRD and YFP-FKBP-DCRD translocated to the membrane upon rapamycin application (*Figure 5D*, *Figure 5—figure supplement 1*). Meanwhile, FRB was also fused onto the channel to construct $\alpha_{1DS}$-$G_{12}$-FRB for inducible dimerization with FKBP-PCRD or FKBP-DCRD, to facilitate peptide-mediated CMI according to the schemes of combination II and III (*Figure 4E*, *Figure 5D*). More potently than the earlier design but in a similar time course (version 1, *Figure 5B, C*), rapamycin induced strong CMI effects on $I_{Ca}$ within 4–5 min (version 2, *Figure 5E,F*). Both CDI and VGA were substantially attenuated: $S_{Ca}$ (0.41 ± 0.06, n = 5) and $I_{peak}$ (35% ± 3%, n = 5). Both indices reached the plateau within 4–5 min, which was about the speed of full solution exchange in our recording system, indicating that rapamycin perfusion was the major time-limiting factor. Thus, the acute nature of CMI was strongly supported by the temporal profiles of inducible CMI effects on CDI/VGA (both version 1 and version 2).

To further consolidate the results, direct drug-channel effects were examined for rapamycin as the negative controls; and full CDI and VGA profiles were characterized across the whole voltage range before and after applying rapamycin (*Figure 5—figure supplement 2*).

## CMI is a unique type of inhibition but sharing similarities with CDI

One interesting feature we discovered from rapamycin-inducible CMI was that the current amplitude at 300 ms ($I_{300}$) stayed at the same levels during the whole time course (*Figure 6A*, top two rows), in contrast to rapidly decayed $I_{peak}$ as outlined by blue dotted lines. In this context, CMI was quite

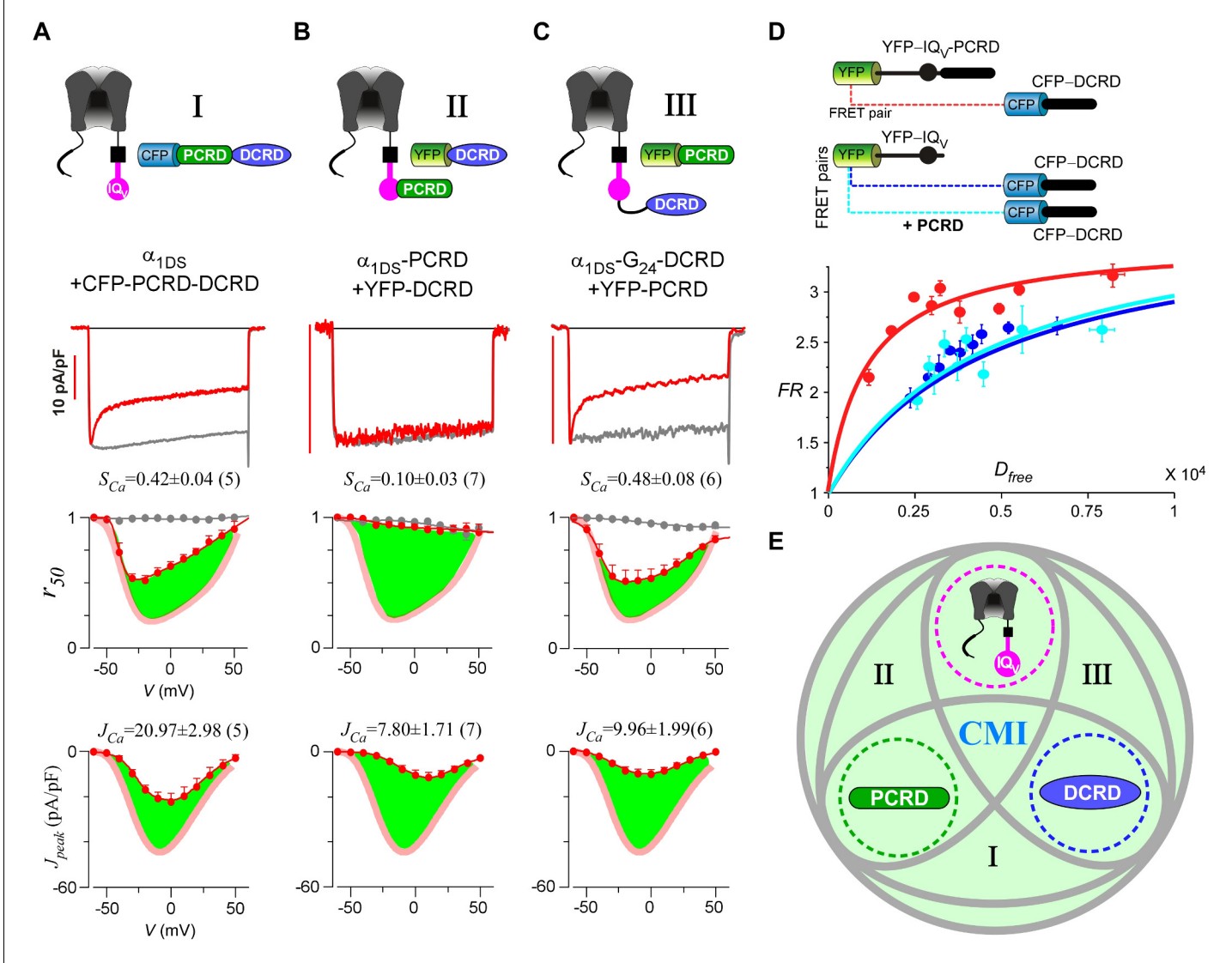

**Figure 4.** Cooperation by PCRD, DCRD and $IQ_V$ to induce CMI. (**A**) Both CDI and VGA of $\alpha_{1DS}$ channels were attenuated by pre-linked PCRD-DCRD illustrated in the scheme (top), shown in the exemplar traces (middle) and voltage-dependent $r_{50}$ and $J_{peak}$ profiles (lower two panels). The potency of CMI was indexed by $S_{Ca}$ and $J_{Ca}$ values and also illustrated by the green areas. Profiles of $\alpha_{1DS}$ control were indicated by thick semitransparent lines in red (lower two panels). (**B**) Both CDI and VGA of $\alpha_{1DS}$-PCRD were strongly prohibited by DCRD. (**C**) Both CDI and VGA of $\alpha_{1DS}$-$G_{24}$-DCRD were attenuated by PCRD. (**D**) FRET 2-hybrid assays demonstrated that pre-linked $IQ_V$-PCRD motif (YFP tagged) exhibited strong binding with CFP-DCRD, with higher binding affinity ($K_d = 1135$, units in donor-cube fluorescence intensity) than the binding affinity ($K_d = 4700$) between CFP-DCRD and YFP-$IQ_V$ itself (without PCRD being fused). Additional PCRD peptides did not make any appreciable change ($K_d = 4624$) for the binding between CFP-DCRD and YFP-$IQ_V$, unable to rescue the low affinity back to the high level that the constitutive PCRD fusion (YFP-$IQ_V$-PCRD) could achieve. $FR_{max}$ values for all these binding curves were similar (3.50—3.87). Each data point represented the $FR$ (y-axis, FRET ratio) and $D_{free}$ (x-axis, free donor concentration) values averaged over five adjacent individual cells sorted by $D_{free}$. (**E**) Working model to illustrate the collaboration among three components of PCRD, DCRD and $IQ_V$ (embedded within $\alpha_{1DS}$) for CMI induction. Grey circles represent the effective combinations, which require that at least two (out of three) components are closely engaged (*e.g.*, fusion) to form the trio complex incorporating the third (separate) component. The three key combinations are I ($IQ_V$+PCRD-DCRD), II ($IQ_V$-PCRD+DCRD) and III ($IQ_V$-DCRD+PCRD), where '–' denotes fusion or spatial closeness and '+' indicates the separate peptide to be coexpressed. In addition, the positive control ($IQ_V$-PCRD-DCRD) also represents the native long variant $\alpha_{1DL}$; and the three components (dotted circles in green, pink and blue) completely separate to each other ($IQ_V$+PCRD+DCRD) produce no CMI effect, serving as one negative control.

The following figure supplements are available for figure 4:

**Figure supplement 1.** Functional similarities between $PCRD_F$ and $PCRD_D$.

*Figure 4 continued*

**Figure supplement 2.** Cooperative perturbation of apoCaM/IQ$_V$ binding by DCRD and PCRD.

unique compared to other conventional inhibitions. In the run-down process (*Kepplinger and Romanin, 2005*) or the blockage by isradipine (*Anekonda and Quinn, 2011*), the number of functional channels was reduced, which resulted into attenuations on $I_{Ca}$ during the whole depolarization step including both $I_{peak}$ and $I_{300}$ (*Figure 6A*, lower two rows). In contrast, $S_{Ca}$ remained constant for $\alpha_{1DS}$ undergoing run-down or blockage since intrinsic gating properties (*e.g.*, CDI) of each channel were not altered, whereas $S_{Ca}$ was attenuated in CMI (*Figure 6—figure supplement 1A*).

Would such constant $I_{300}$ be just a coincidence? By revisiting different channel variants tested earlier, we concluded that $I_{300}$ indeed remained at similar amplitudes for them all (*Figure 6B*), whereas their $I_{peak}$ and $S_{Ca}$ exhibited broad dynamic ranges. To confirm, depolarization pulses of longer duration (1000 ms) were also utilized to conduct constitutive and rapamycin-inducible CMI (*Figure 6—figure supplement 2*). In both modalities of CMI, the behavior of $I_{300}$ was similar to $I_{1000}$ and both indices remained constant. In addition, CDI already approached its steady-state at 300 ms (end-stage CDI), and the slight difference between $I_{300}$ and $I_{1000}$ was largely due to VDI and negligible in this study. Throughout this work, $I_{300}$ was employed as the major index to for both CMI and CDI analyses.

The constant $I_{300}$ (or $J_{300}$) essentially set the lower limit for CMI. That said, while CMI potency was getting higher, $J_{peak}$ was tuned down to more closely approach $J_{300}$ (*e.g.*, *Figure 6B vs Figure 1B*). More clear evidence came from rapamycin-inducible CMI: the attenuated $I_{peak}$ and constant $I_{300}$ altogether account for $S_{Ca}$ (CDI) attenuation; and for ultrastrong CMI, $S_{Ca}$ would be nearly abolished (0), indicating that $I_{peak}$ would reach approximately the level of its lower limit ($I_{300}$).

These lines of evidence also led us to conclude that a reduction of $Ca^{2+}$ influx is guaranteed for CMI, despite concurrent VGA/CDI (activation/inactivation) attenuations. Such concurrency apparently would cause contradictory effects on $Ca^{2+}$ influx: inhibition of CDI (ICDI) would tend to enhance $Ca^{2+}$ influx, in opposition to attenuation on VGA (less $Ca^{2+}$ influx). Even though, we just clarified that CMI attenuation on CDI is actually realized by reducing $I_{peak}$ while maintaining the end-stage $I_{Ca}$ (indexed with $I_{300}$ or $I_{1000}$), which ensures the overall $Ca^{2+}$ influx is reduced. Thus, the major uncertainty was relieved for CMI to emerge as a one new modality of $Ca^{2+}$ channel inhibition.

These observations are consistent with the speculation that CDI might be the reversed process to the apoCaM promotion of open probability (*Adams et al., 2014*). High similarities are shared by DCT/apoCaM-dependent CMI and $Ca^{2+}$/CaM-mediated CDI: functionally CMI (ultrastrong) and CDI (end-stage) could result into similar gating; and mechanistically they appear to be triggered by similar events: the pre-association between apoCaM and IQ$_V$ is either totally abolished (CMI) or drastically altered (CDI) (*Figure 6C*). These facts suggest that potentially the same set of 'core machinery' (cyan square) mediates both CMI and CDI with very similar structures and structural changes. To exclude potential complications from ambient $Ca^{2+}$/CaM, we performed FRET experiments to ensure that the interactions within the trio complex of IQ$_V$/PCRD/DCRD were largely unaffected by $Ca^{2+}$/CaM produced from ionomycin-introduced $Ca^{2+}$ and endogenous CaM, although $Ca^{2+}$/CaM indeed exhibited higher binding affinity to IQ$_V$ than apoCaM (*Figure 6—figure supplement 3*).

## CMI inhibits Ca$_V$1.3-mediated oscillation and pacemaking in SNc neurons

Ca$_V$1.3 plays a pivotal role in subthreshold oscillation and suprathreshold pacemaking in diverse cell types including neurons (*Chay and Keizer, 1983*; *Comunanza et al., 2010*), *e.g.*, dopaminergic neurons in the substantia nigra compacta (SNc) (*Chan et al., 2007*). Pathophysiological linkages of PD or other neurodegenerative diseases with Ca$_V$1.3 functions have been evidenced in multiple lines of studies (*Guzman et al., 2009*; *Puopolo et al., 2007*), including Ca$_V$1.3 antagonists as potential therapeutic interventions (*Anekonda and Quinn, 2011*; *Pasternak et al., 2012*; *Triggle, 2007*). To explore the physiological tuning of CMI and its therapeutic potentials, we constructed a customized model for SNc neuron and Ca$_V$1.3 channels using the software Neuron (*Carnevale and Hines, 2006*). Computational analyses were performed to compare the effects between different levels of

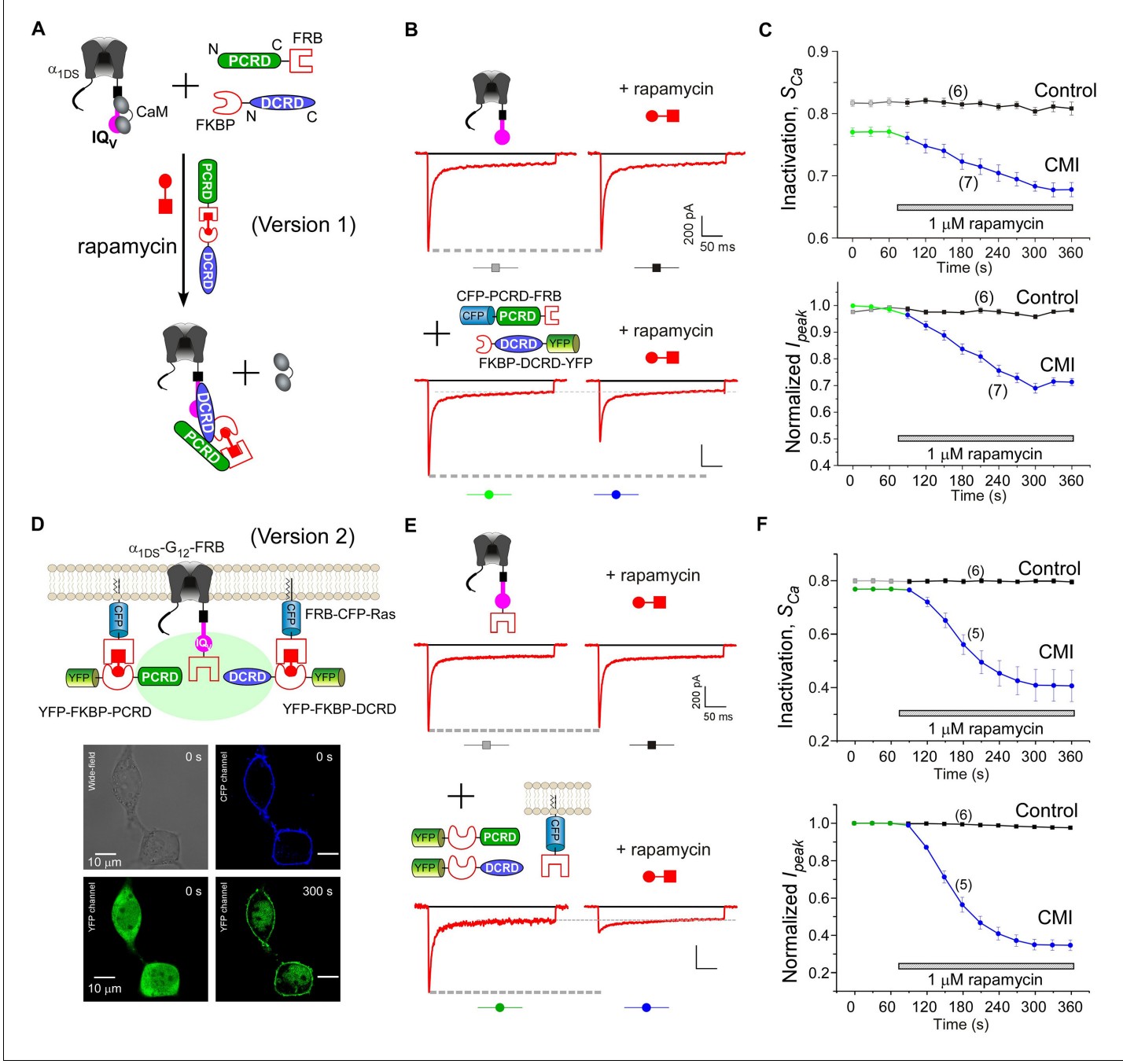

**Figure 5.** Design schemes and experimental implementations of rapamycin-inducible CMI. (**A**) Design principles for chemical-inducible CMI. Rapamycin simultaneously binds one FKBP and one FRB to combine any two FRB/FKBP-tagged components (PCRD and DCRD, in this version 1) selected from the three components of PCRD, DCRD and $IQ_V$. Thus, the three components would form the combinations (combination I, in the version 1) to satisfy the requirement of cooperative CMI (*Figure 4E*). (**B**) One representative implementation of chemical-inducible CMI. According to the design of version one in (**A**), PCRD-FRB and FKBP-DCRD were constructed and coexpressed with $\alpha_{1DS}$. Exemplars of $Ca^{2+}$ current traces demonstrated that acute effects were induced by applying 1 μM rapamycin to the multi-component system of version 1 (lower panel), in contrast to $\alpha_{1DS}$ control with no appreciable changes in $I_{Ca}$ (upper panel). (**C**) Statistical summary of rapamycin-induced CMI (version 1). Averaged values over multiple cells (number indicated in parentheses) for the indices $S_{Ca}$ (upper) or normalized $I_{peak}$ (lower) demonstrated time-dependent attenuations on both CDI and VGA upon rapamycin application as compared to the control group. (**D**) Rapamycin perfusion induced rapid translocation of YFP-FKBP-DCRD or YFP-FKBP-PCRD (version 2) onto the membrane by linking with FRB-CFP-Ras within 5 min, shown by confocal images via wide-field, CFP, and YFP channels. The local concentrations of YFP-FKBP-DCRD and YFP-FKBP-PCRD were substantially enhanced as suggested by the condensed YFP fluorescence at the membrane (outlined by CFP fluorescence). (**E** and **F**) According to rapamycin-inducible CMI of version 2, recombinant channels of $\alpha_{1DS}$-$G_{12}$-FRB were

*Figure 5 continued on next page*

*Figure 5 continued*

coexpressed with cytosolic YFP-FKBP-PCRD and YFP-FKBP-DCRD. To enhance local concentrations of FKBP-tagged peptides near the channels, membrane-localized FRB-CFP-Ras was also overexpressed. Exemplars of $Ca^{2+}$ current traces (E) exhibited strong attenuation (lower), in contrast to stable $I_{Ca}$ from $\alpha_{1DS}$-$G_{12}$-FRB alone (upper). In contrast to the control group, temporal profiles of rapamycin-inducible CMI (version 2) indicated strong attenuations on CDI ($S_{Ca}$, upper) and VGA (normalized $I_{peak}$, lower). Based on totally five trials (cells), $S_{Ca}$ changed from $0.77 \pm 0.00$ to $0.41 \pm 0.06$ and $I_{peak}$ was reduced to $35\% \pm 3\%$ of the basal levels (F).

The following figure supplements are available for figure 5:

**Figure supplement 1.** Temporal profiles of fluorescent images for rapamycin-induced membrane-targeting.

**Figure supplement 2.** Detailed characterizations for rapamycin-induced CMI.

CMI attenuations, to simulate the tuning of CMI by multiple factors including endogenous DCTs and alternative splicing, apoCaM fluctuations, and apoCaM buffer proteins such as calpacitin (*Gerendasy, 1999*; *Xia and Storm, 2005*). Under CMI of different potency, $I_{Ca}$ from $\alpha_{1DS}$ (no CMI), $\alpha_{1DL}$ (intermediate CMI), or $\alpha_{1DS}$ with inducible peptide dimerization (ultrastrong CMI), exhibited distinct $I_{peak}$, $S_{Ca}$ and $Ca^{2+}$ influx (*Figure 7A—C*, left) (green areas indicating the reduction of $Ca^{2+}$ influx). Meanwhile, the time rates of $Ca^{2+}$ oscillation and autonomous spiking in the SNc model were also tuned down to different levels in accordance with $I_{Ca}$ attenuations or CMI potencies (*Figure 7A—C*). Experimentally, CMI was compared with other conventional modalities of channel inhibition, for their differences in $I_{Ca}$ attenuations (*Figure 6A*, *Figure 6—figure supplement 1*). In parallel, we also simulated the conventional blockage in the $Ca_V1.3$/SNc model (*Figure 7D*). Similar attenuations in oscillation and pacemaking were observed from the SNc model when the moderate (28%) blockade clearly reduced overall $Ca^{2+}$ influx (green area), supporting the current strategy to develop PD therapeutics based on conventional $Ca_V1.3$ antagonists (*Gudala et al., 2015*; *Hurley et al., 2013*; *Kang et al., 2012*; *Pasternak et al., 2012*).

Collectively, one common rule was deduced for CMI, CDI and other modalities of inhibition: $Ca^{2+}$ influx via $Ca_V1.3$ serves as the major index to determine the downstream of oscillation/pacemaking, *i.e.*, reduction of $Ca^{2+}$ influx leads to attenuation of pacemaking whereas enhancement of $Ca^{2+}$ influx causes augmentation of pacemaking. These analyses on the potential roles of CMI together with those on CMI mechanisms (*Figure 6C*) suggest that multiple molecular processes in CMI, especially those pertaining to apoCaM and the trio complex, are crucial for excitability and signaling of SNc neurons as well as related pathology, such as PD (*Hurley et al., 2013*; *Scharinger et al., 2015*) (*Figure 7E*). In this context, CMI, emerging as one key regulatory mechanism distinct from others, essentially reduces $Ca^{2+}$ influx, local and cytosolic $Ca^{2+}$ concentration, and oscillation and pacemaking, all of which are potentially subject to bidirectional tuning under pathophysiological conditions.

## Discussion

This study focuses on the principle of CMI mediated by multiple carboxyl-tail motifs to compete apoCaM off the channel. Such cooperative competition is applied to rapamycin-inducible inhibition on $Ca_V1.3$ channels, unveiling that the channel inhibited by ultrastrong CMI should be functionally equivalent to the channel in end-stage CDI. Furthermore, by acute and cooperative inhibition of $Ca^{2+}$ influx, this unique CMI serves as potential therapeutic interventions, as demonstrated by attenuation of $Ca_V1.3$-dependent pacemaking with a computational model of SNc neurons.

### CMI across LTCC family and potential physiological linkages

For LTCCs, gating inhibition by carboxyl-tail motifs (CMI) seems to be a universal mechanism preventing excessive $Ca^{2+}$ influx and intracellular $Ca^{2+}$ overload. In $Ca_V1.1$ and $Ca_V1.2$ channels, C-termini of $\alpha_1$ subunits are subject to proteolytic cleavage and the site of cleavage is located between PCRD and DCRD (*Abele and Yang, 2012*; *Hulme et al., 2005*, *2006*). Hence, CMI serves as the potential mechanism for endogenous DCRD-containing fragments to attenuate $I_{Ca}$ by cooperating with PCRD and $IQ_V$. For $Ca_V1.3$, the possibility of proteolytic cleavage has not yet been fully explored. Instead, DCT is subject to alternative splicing thus generating $\alpha_{1D}$ isoforms in two major

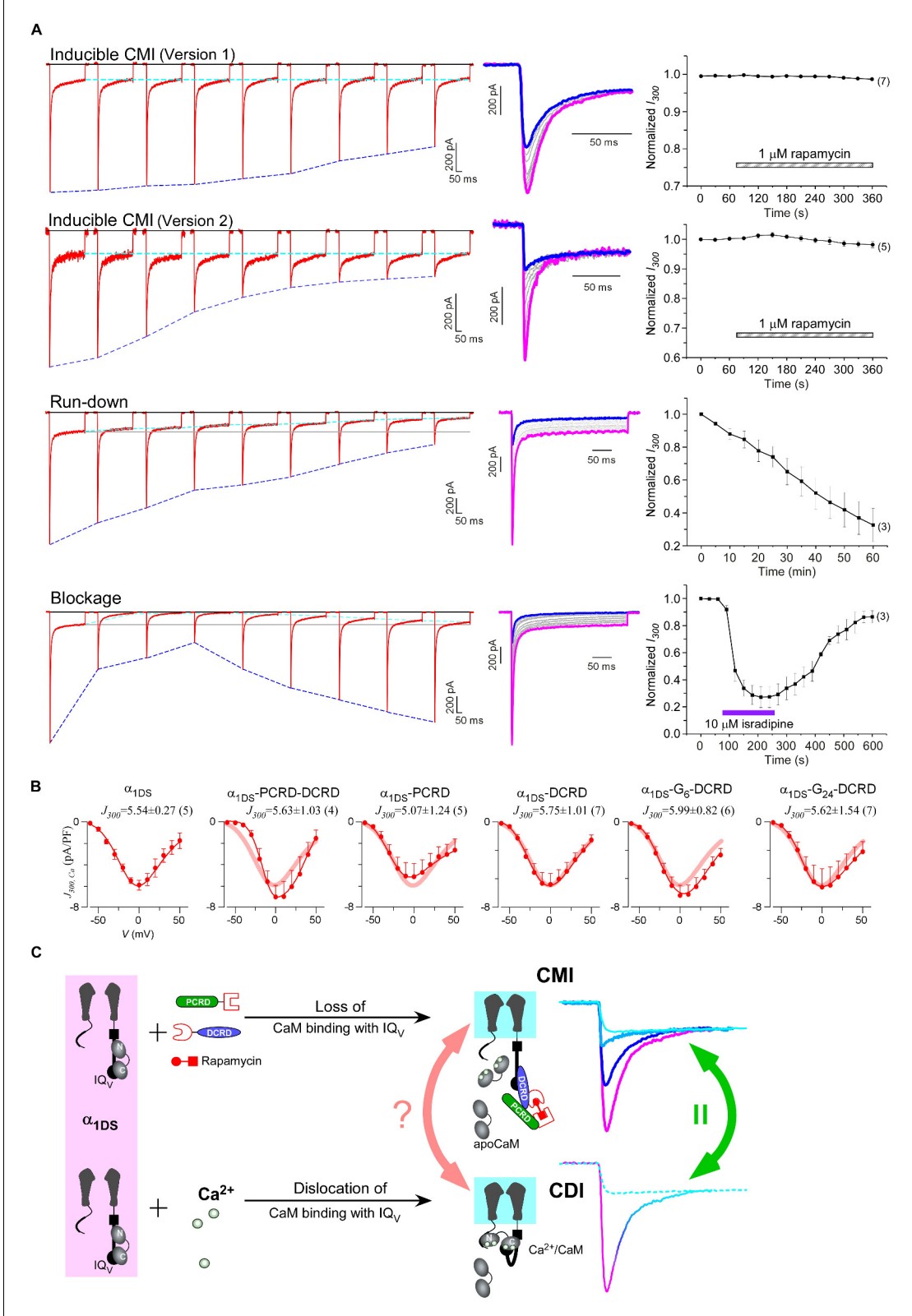

**Figure 6.** Mechanistic insights enlightened by unique features of CMI. (**A**) Temporal profiles between CMI and conventional inhibitions were compared. For rapamycin-inducible CMI (two versions in the top two rows of panels), representative $Ca^{2+}$ current traces in rapamycin were selected from sequential time-points (left column) and superimposed together (middle column) for comparison. The time sequence was indicated by color: the first in pink, intermediate in grey, and the last in blue. $I_{peak}$ exhibited the trend of inhibition but not for $I_{300}$ ($I_{Ca}$ at 300 ms) (right column). In contrast, for run-

*Figure 6 continued on next page*

*Figure 6 continued*

down process and isradipine blockage (bottom two rows), both $I_{peak}$ and $I_{300}$ exhibited the declining trends indicating substantial inhibitions. (B) $Ca^{2+}$ current density at 300 ms ($J_{300,Ca}$) remained at the same level (indicated by the $J_{300}$ values at −10 mV), for various channel variants under test, regardless of whether CMI was in effect. Thick lines (semitransparent in red) represent the $J_{300,Ca}$ profile of $\alpha_{1DS}$ control. (C) Functional and the structural insights into the two modes of inhibition. DCT/apoCaM-dependent CMI and $Ca^{2+}$/CaM-mediated CDI result in indistinguishable gating (green arrows) appearing due to similar causes, *i.e.*, either total or partial loss of the apo-state CaM/$IQ_V$ complex. That says, in CMI, apoCaM pre-association is totally lost; and in CDI, apoCaM is calcified and dislocated from the pre-association sites. Although the triggers are different for CMI *vs* CDI, *i.e.*, DCT competing (off apoCaM) *vs* $Ca^{2+}$ binding (onto apoCaM), the similarities between the two different inhibitory regulations of CMI and CDI invite the hypothesis that the core gating machinery (cyan squares) upon depolarization might step into the same scenario/mode including structural details (red arrows). A series of current traces (on the right) indicate CMI with different potency (enhanced from pink to cyan, upper), in comparison with the trace at different stages of CDI (developed from pink to cyan, lower) superimposed with the trace from ultrastrong CMI (dotted trace in cyan). These analyses point to one important notion that the lower limit of CMI is determined by end-stage CDI (the traces or the phases in cyan), and thus the dynamic changes for CMI effects on $\alpha_{1DS}$ (indexed with $I_{peak}$ and $S_{Ca}$) are preset, *i.e.*, from the pink (no CMI and ultrastrong CDI) to the cyan (ultrastrong CMI and no CDI).

The following figure supplements are available for figure 6:

**Figure supplement 1.** Detailed comparisons among CMI, run-down and blockage.

**Figure supplement 2.** Indices at different time points of $I_{Ca}$ to quantify CMI and end-stage CDI.

**Figure supplement 3.** The high-affinity binding between DCRD and the channel was not perturbed by $Ca^{2+}$/CaM.

categories: with ($\alpha_{1DL}$) or without ($\alpha_{1DS}$ or similar variants) autonomous CMI (*Bock et al., 2011*; *Singh et al., 2008*), both broadly expressed in various tissues and organs. $Ca_V1.4$ channels have profound CMI due to its strong DCT against apoCaM, as evidenced by weak inactivation and small currents in native cells, altogether allowing sustained $Ca^{2+}$ influx and continuous neural transmission in retinal neurons (*Singh et al., 2006*) and immune cells (*Omilusik et al., 2011*). To this end, CMI is a prevalent modality of inhibition across $Ca_V1$ family members, awaiting further investigations into the molecular mechanisms and pathophysiology.

## Mechanistic insights into $Ca_V1$ gating enlightened by CMI

Based on constitutive and inducible CMI effects, we have gained considerable insights into $Ca_V1$ gating. First of all, CMI as an emerging modality of channel inhibition resembles CDI, one of the prominent features of $Ca^{2+}$ channels (*Ben-Johny and Yue, 2014*). Targeting the mechanisms of CDI, most structural efforts have been directed to channels in $Ca^{2+}$ conditions, evidenced by multiple studies focusing on $Ca^{2+}$/CaM bound with key channel motifs such as IQ or $IQ_V$ (*Ben Johny et al., 2013*; *Fallon et al., 2005*; *Kim et al., 2008*; *Mori et al., 2008*) and NSCaTE (*Dick et al., 2008*), which have not achieved unequivocal viewpoints toward CDI mechanisms. CMI emphasizes on the importance of apoCaM-bound and apoCaM-off structures in $Ca^{2+}$-free conditions, which represent the states of 'Activated' (pre-CMI) and 'Inhibited' (post-CMI), promisingly holding the key to understand the core mechanisms that control the gating of $Ca^{2+}$ channels. Functionally, CMI helps clarify the long-standing complication on the overall effects of CDI on the actual $Ca^{2+}$ influx into the cell. Historically, CDI is considered as a negative feedback to autonomously reduce $Ca^{2+}$ entries (*Alseikhan et al., 2002*; *Budde et al., 2002*; *Liu et al., 2010*). On the other hand, stronger CDI is often accompanied with larger amplitude or facilitated activation of $I_{Ca}$ (*Adams et al., 2014*; *Liu et al., 2016*; *Singh et al., 2008*), altogether raising the question whether $Ca^{2+}$ influx is reduced or enhanced by CDI. Here we provide direct evidence that attenuation of CDI via endogenous factors and mechanisms is essentially a reduction of $Ca^{2+}$ influx, assured by the fixed level of $Ca^{2+}$ current at the late phase of $I_{Ca}$, *i.e.*, $I_{300}$ measured in this work, remaining the same regardless of the changes in $I_{peak}$ or $S_{Ca}$. In this context, CMI (in inverse correlation to CDI strength) is the more direct index of channel inhibition. In contrast, CDI strength itself ($S_{Ca}$) could be changed by various mechanisms of action besides [apoCaM] tuning, which may or may not actually reduce $Ca^{2+}$ influx. For instance, channel mutations in congenital diseases, *e.g.*, Timothy syndrome (*Splawski et al., 2004*), could cause less inactivation in $I_{Ca}$ and thus elevate the $Ca^{2+}$ entries. Also, small-molecule compounds could produce confounding effects on $Ca^{2+}$ influx by triggering multiple events opposed to

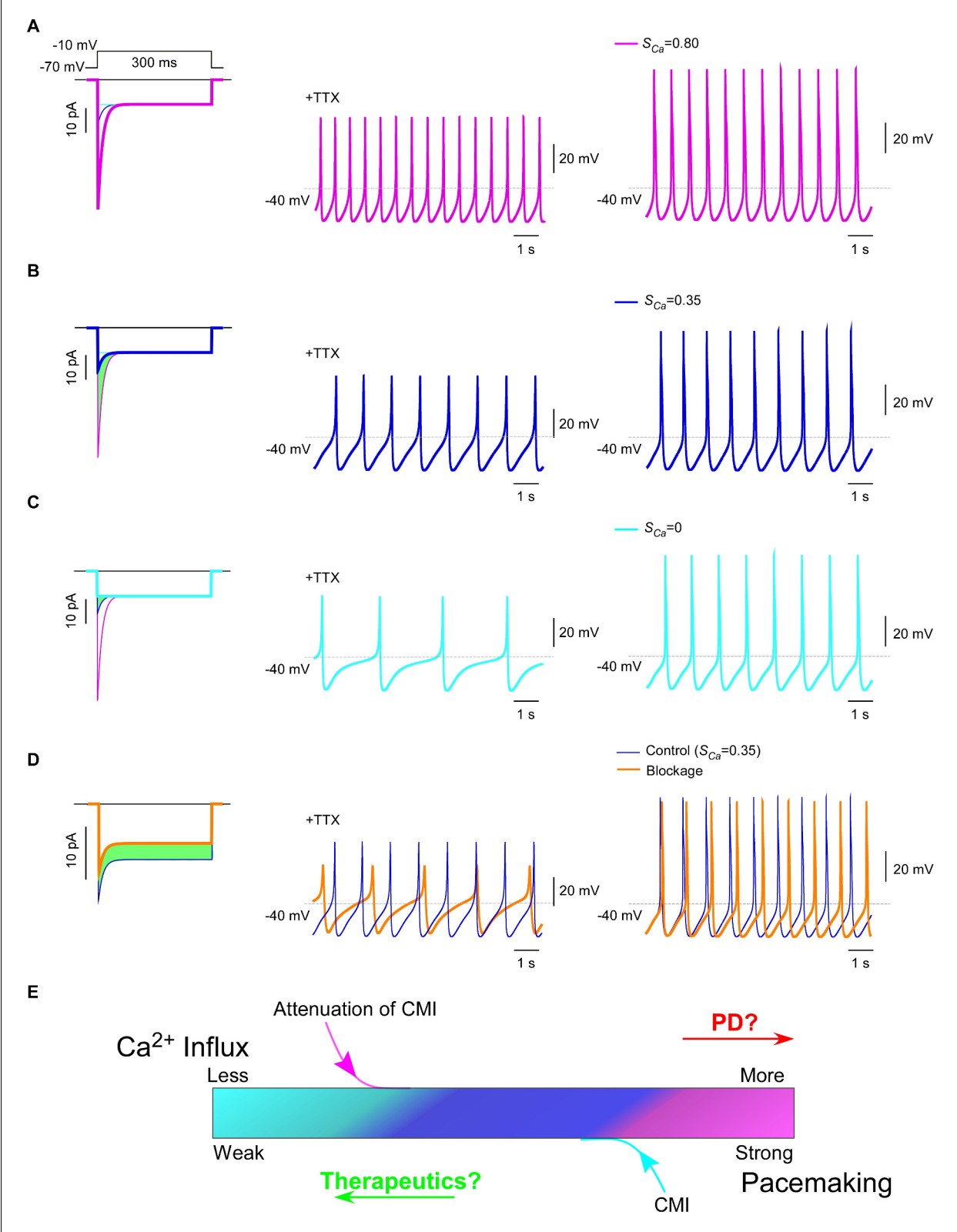

**Figure 7.** Inhibition of SNc oscillation and pacemaking by CMI. (**A** and **B**) Oscillation and pacemaking of SNc neurons are mediated by $Ca_V1.3$, presumably mixed with both short ($\alpha_{1DS}$) and long ($\alpha_{1DL}$) splice variants. Lack of CMI, $\alpha_{1DS}$ (pink) exhibited large $I_{peak}$ and strong $S_{Ca}$ (**A**), altogether resulted into more $Ca^{2+}$ influx than $\alpha_{1DL}$ (blue), as indicated by the green area (**B**). Side-by-side comparison demonstrated that $Ca_V1.3$-dependent oscillation (middle column) and pacemaking (right column) were slowed down, presumably by intrinsic CMI of $\alpha_{1DL}$ in comparison to $\alpha_{1DS}$. (**C**) Effects of

*Figure 7 continued on next page*

Figure 7 continued

ultrastrong CMI (cyan) on SNc neurons. Under the CMI of the maximum potency (left), $I_{peak}$ was reduced to the lower limit, *i.e.*, the level of $I_{300}$, and $S_{Ca}$ was approaching 0. Accompanied with reduction in $Ca^{2+}$ influx (green area), oscillation (middle) and pacemaking (right) rates were also further reduced compared with $\alpha_{1DS}$ (A) or $\alpha_{1DL}$ (B). (D) Effects of conventional blockage. Conventional inhibition by blockers (orange) attenuated $I_{Ca}$ (28% reduction) and $Ca^{2+}$ influx (green area) but kept $S_{Ca}$ unaltered (left), by which oscillation (middle) and pacemaking (right) rates were slowed down compared with the $\alpha_{1DL}$ control (B). (E) The potential regulatory scheme for $Ca_V1.3$-mediated pacemaking in SNc neurons. The actual gating of $I_{Ca}$ (similar to B) could fall into intermediate levels between two extreme conditions of rather weak CMI (apoCaM-bound and capable of strong CDI) (A), or ultrastrong CMI (apoCaM-off and abolishment of CDI) (C). Hence, both $Ca_V1.3$ and pacemaking behaviors are potentially bidirectionally regulated, *e.g.*, apoCaM and apoCaM-binding proteins, to maintain the homeostatic balance of $Ca^{2+}$ influx. Pathological dysregulations of $Ca_V1.3$ and the resulted $Ca^{2+}$ imbalance might underlie PD and other diseases, as the rational to develop therapeutic perturbations based on CMI.

each other (*Hille, 2001*), *e.g.*, roscovintine (*Yarotskyy et al., 2010*) reportedly damps activation and meanwhile enhances inactivation, raising uncertainties in its actual effects on $Ca^{2+}$ influx.

## Therapeutic potentials of CMI-based interventions for PD and beyond

$Ca_V1.3$ acts as the dominant factor underlying subthreshold oscillation and pacemaking activities in SNc dopaminergic neurons, although the detailed mechanisms are not fully elucidated (*Chan et al., 2007*; *Durante et al., 2004*; *Putzier et al., 2009*). It is currently believed that activity-dependent engagement of $Ca_V1$ channels elevates mitochondrial oxidant stress and vulnerability of SNc neurons, contributing to disease progression of PD (*Guzman et al., 2010*). Additional support also comes from clinical evidence that $Ca_V1$ blockers for hypertension treatment, *e.g.*, 1,4-dihydropyridines (DHPs) appear to reduce the risk of PD (*Pasternak et al., 2012*; *Ritz et al., 2009*). However, concerns are raised on developing PD therapeutics based on DHP or like inhibitors. First of all, it has been argued whether $Ca_V1.3$ is indeed a prerequisite for pacemaking and autonomous oscillation, because of pharmacological complications of DHP, including non-specific side effects and discrepancies in dose dependence (*Guzman et al., 2009*; *Putzier et al., 2009*). Viral or transgenic delivery of CMI peptides to $Ca_V1.3$ channels in neurons could circumvent above problems, in hope to unequivocally confirm the actual role of $Ca_V1.3$. Secondly, PD interventions deploying $Ca_V1.3$ antagonists could be improved in the following aspects: (1) specificity, for which genetically-encoded CMI-based inhibitors provide the desired specificity intrinsic to $Ca_V1$ channels; (2) assurance of $Ca^{2+}$ reduction, for which CMI is mechanistically guaranteed to reduce $Ca^{2+}$ influx; however, certain antagonists may not be able to provide such assurance due to aforementioned reasons.

CMI-based interventions and therapeutics could be beneficial to other diseases in addition to PD. A wide spectrum of mental disorders are closely involved in dysregulations of $Ca_V1$ channels, in that disease-linked mutations in $\alpha_1$ and $\beta_2$ subunits result into abnormal gating properties (*Azizan et al., 2013*; *Cross-Disorder Group of the Psychiatric Genomics Consortium, 2013*; *Schizophrenia Working Group of the Psychiatric Genomics Consortium, 2014*; *Scholl et al., 2013*). Encouraged by the results from SNc modeling that the dysregulated $Ca_V1.3$, $Ca^{2+}$ influx and pacemaking could be corrected back to normal, we expect CMI and its therapeutic potentials to manifest in diverse pathophysiology including $Ca_V1$ channelopathies.

## Historical context and future directions

DCT and DCRD effects have been suggested in a few prior works, *e.g.*, DCT peptides truncated from long variants of $Ca_V1.3$ or $Ca_V1.4$ to coexpress with the truncated channels lacking the DCT motif (*Singh et al., 2008*, *2006*), mainly to demonstrate the ICDI (inhibition of CDI) effect. Here, to advance the understanding, we clarify that multiple C-terminal motifs cooperatively and acutely compete apoCaM off the channel; and unveil that CMI functionally resembles CDI such that the maximum potency of CMI is preset by end-stage CDI, which also ensures CMI reduces the overall $Ca^{2+}$ influx. Moreover, as one major goal beyond basic biophysics, we achieve CMI-based peptides for $Ca_V1.3$ of native forms, including the short splice variant $\alpha_{1DS}$ in this work and potentially also the long variant $\alpha_{1DL}$, another major isoform in native tissues (*Yang et al., 2015*). In parallel, one study focusing on CaM, the other side of the competition, speculated on CDI mechanisms as a simple CaM-off process based on the data produced with local enrichment of apoCaM (*Adams et al., 2014*), which, in our view, may have the follow concerns. First, prescreening is needed to control the

baseline gating since CaM itself is very well expressed in cells and easily reaches concentrations high enough to diminish the dynamic space to gauge any further change in CDI (*Liu et al., 2010*). Second, even with above practical issues being handled, any claims about CDI still await further proof with direct evidence based on actual inhibition such as CMI. Third, apoCaM, due to its central position and multifaceted roles in cell signaling (*Cheung, 1980*; *Chin and Means, 2000*; *Guzman et al., 2010*; *Hoeflich and Ikura, 2002*), is nearly impossible to develop into molecular tools and therapeutics.

To expand CMI-based inhibitors onto in vivo applications, the approach of rapamycin-mediated heterodimerization may not be directly applicable due to potential interferences with normal cell proliferation, growth and survival (*MacMillan, 2013*). Nevertheless, the proof-of-concept prototypes in this work lay the foundations for further development and optimization, *e.g.*, by way of light-inducible dimerization (*Kennedy et al., 2010*; *Levskaya et al., 2009*; *Yazawa et al., 2009*), to explore the biological ramifications and therapeutic potentials of CMI in diverse settings.

# Materials and methods

## Molecular biology

$\alpha_{1D\_Short}$ (denoted as $\alpha_{1DS}$) variants were constructed by introducing a unique XbaI site following the IQ domain. PCRD was cloned from either Ca$_V$1.4 $\alpha_{1F}$ (NP005174, GenBank accession number, the same as follows) or Ca$_V$1.3 $\alpha_{1D}$ (NM_000720); and DCRD was based on Ca$_V$1.4 $\alpha_{1F}$ (NP005174), and CFP/YFP-tagged DCRD constructs were made using a similar process as described previously (*Liu et al., 2010*). Then other CFP/YFP-tagged constructs were cloned by replacing DCRD with appropriate PCR amplified segments, via unique NotI and XbaI sites, including YFP-PCRD, CFP-PCRD-DCRD, and CFP-CaM. Peptides of IQ$_V$ and IQ$_V$-PCRD were cloned from Ca$_V$1.2 $\alpha_{1C}$ (NM_199460.3) and Ca$_V$1.3 $\alpha_{1D}$ (NM_000720), based on preIQ$_3$-IQ and preIQ$_3$-IQ-PCRD as the templates respectively. YFP-IQ$_V$ and YFP-IQ$_V$-PCRD were cloned similarly as YFP-PCRD. Segments of PCRD-DCRD, PCRD, DCRD with different glycines (G$_0$, G$_6$ and G$_{24}$) were PCR-amplified with SpeI and XbaI sites and inserted into aforementioned $\alpha_{1DS}$.BSCaM$_{IQ}$, serving as the apoCaM buffer in this study, was kindly provided by Dr. A. Persechini (*Black et al., 2006*). Rapamycin-inducible system consisted of FRB, which was based on 93 a.a. rapamycin binding motif of MTOR; and FKBP, which was 108 a.a. human FKBP-12 (AAA58472). Constructs of CFP-PCRD-FRB, YFP-FKBP-DCRD, YFP-FKBP-PCRD and FRB-CFP-ras were made by appropriate design. PCRD segment was amplified by PCR with flanking NotI and EcoRI then cloned directionally via these two unique sites into CFP-FRB_pcDNA4_HisMaxC expressing plasmids. DCRD segment was amplified by PCR with flanking BamHI and EcoRI then cloned directionally via these two unique sites into YFP-FKBP_pcDNA4_His-MaxC expressing plasmids. The fusion of YFP-FKBP-PCRD and FRB-CFP-ras were ligated by overlap extension PCR with flanking KpnI and XbaI then cloned into pcDNA3 expressing plasmids via these two unique sites. To make constructs of PCRD-FRB and FKBP-DCRD, segments from CFP-PCRD-FRB and YFP-FKBP-DCRD respectively were amplified by PCR with flanking KpnI and XbaI then cloned directionally into pcDNA4 vector. For chimeric channel $\alpha_{1DS}$-G$_{12}$-FRB, a linker containing 12 glycine residues was fused with FRB by overlap PCR with flanking SpeI and XbaI and cloned into $\alpha_{1DS}$ containing engineered cloning sites before the stop codon.

## Transfection of cDNA constructs

For whole-cell electrophysiology and confocal fluorescence imaging, HEK293 cells were cultured in 60 mm dishes or 35 mm No. 0 glass-bottom dishes, and constructs were transiently transfected according to an established calcium phosphate protocol (*Liu et al., 2010*). HEK293 cell line was generously provided by Dr. Zhijie Chang (Tsinghua University). The cell line was free of mycoplasma contamination, checked by PCR with primers 5'- GGCGAATGGGGTGAGTAACACG −3' and 5'- CGGA TAACGCTTGCGACCTATG −3'. 5 μg of cDNA encoding the desired $\alpha_{1D}$ subunit, along with 4 μg of rat brain $\beta_{2a}$ (M80545) and 4 μg of rat brain $\alpha_2\delta$ (NM012919.2) subunits were applied. All of the above cDNA constructs contained a cytomegalovirus promoter. To enhance expression, cDNA for simian virus 40 T antigen (1–2 μg) was also co-transfected with channel constructs. For all the peptides co-transfected, 2 μg of plasmids were added together with channels. Cells were washed with PBS 6–8 hr after transfection and maintained in supplemented DMEM, then incubated for at least 48

hr in a water-saturated 5% $CO_2$ incubator at 37°C before electrophysiology experiments and confocal microscopy experiments.

For FRET optical imaging, HEK293 cells cultured on 35 mm No. 0 glass-bottom dishes were transfected with cDNAs (1–5 μg each) by Lipofectamine 2000 (Invitrogen, Waltham, MA). Cells were washed with PBS 4–6 hr after transfection and maintained in supplemented DMEM, then incubated for 24–48 hr in a water-saturated 5% $CO_2$ incubator at 37°C before FRET 2-hybrid experiments.

## Whole-cell electrophysiology

Whole-cell recordings of transfected HEK293 cells were performed at room temperature (25°C) using an Axopatch 200B amplifier (Axon Instruments, Sunnyvale, CA). Electrodes were pulled with borosilicate glass capillaries by a programmable puller (P-1000, Sutter Instruments, Novato, CA) and heat-polished by a microforge (MF-830, Narishige, Japan), resulting in 1–3 MΩ resistances, before series resistance compensation of about 70%. The internal solutions contained (in mM): $CsMeSO_3$, 135; $CsCl_2$, 5; $MgCl_2$, 1; MgATP, 4; HEPES, 5; and EGTA, 5, adjusted to 290 mOsm with glucose and pH 7.3 with CsOH. The extracellular solutions contained (in mM): $TEA-MeSO_3$, 140; HEPES, 10; $CaCl_2$ or $BaCl_2$, 10, adjusted to 300 mOsm with glucose and pH 7.3 with TEAOH, all according to the previous reports (*Liu et al., 2016*, *2010*). For run-down experiments, the content of MgATP (Sigma-Aldrich, St. Louis, MO) in the internal solutions was intentionally reduced. Chemical reagents used for blockage experiments (isradipine, Sigma-Aldrich) and drug-inducible experiments (rapamycin, Fisher Scientific, Waltham, MA) were dissolved in DMSO as 10 mM or 1 mM stock solution, stored at −20°C, and then diluted to 10 μM or 1 μM using extracellular $Ca^{2+}$ solution right before experiments. Whole-cell currents were generated from a family of step depolarization (−70 to +50 mV from a holding potential of −70 mV) or a series of repeated step depolarization (−10 mV from a holding potential of −70 mV). Currents were recorded at 2 kHz low-pass filtering of the instrument. Traces were acquired at a minimum repetition interval of 30 s. P/8 leak subtraction was used throughout.

## FRET optical imaging

FRET 2-hybrid imaging experiments were performed with an inverted microscope (Ti-U, Nikon, Japan) with Neo sCMOS camera (Andor Technology, UK). The light source was from the mercury lamp filtered at the appropriate wavelengths for CFP and YFP by the optical filters mounted at the computer-controlled filter wheel (Sutter Instrument) for excitation, subsequently passing the dichroic mirror and the emission filters. Operations and measurements were controlled by the iQ software (Andor Technology). FRET data were acquired and analyzed by an intensity-based two-hybrid assay ($3^3$-FRET) as described (*Butz et al., 2016*; *Erickson et al., 2001*; *Liu et al., 2010*), based on

$$FR = 1 + \frac{FR_{max} - 1}{1 + \frac{K_d}{D_{free}}}$$

where $FR_{max}$ represents the maximum FRET ratio $FR$ pertaining to the receptor (YFP), and $D_{free}$ denotes the equivalent free donor (CFP-tagged) concentration. By fitting the curve of $FR - D_{free}$ with a set of customized Matlab (Mathworks, Natick, MA) codes to iteratively estimate $FR_{max}$ and $D_{free}$, effective dissociation equilibrium constant ($K_d$) can be achieved for each binding pair to evaluate the (relative) affinity. During imaging, the bath solution was Tyrode's buffer, containing (in mM): NaCl, 129; KCl, 5; $CaCl_2$, 2; $MgCl_2$, 1; HEPES, 25; glucose, 30, 300 mOsm, adjusted with glucose and pH 7.3, adjusted with NaOH. Ionomycin (Sigma-Aldrich) was dissolved in DMSO as 1 mM stock solution, stored at −20°C, and diluted to 1 μM using Tyrode's buffer immediately before applications.

## Confocal fluorescence imaging

Fluorescence images were achieved in HEK293 cells transfected with membrane-localized CFP-tagged FRB and cytosolic FKBP-PCRD/DCRD tagged with YFP. Dynamic translocations were observed with a ZEISS (Germany) Laser Scanning Confocal Microscope (LSM710) through a 100X oil objective and analyzed with ZEN 2009 Light Edition software and Adobe Photoshop CS5 software (Adobe Systems, San Jose, CA).

## Computer simulation

Simulations were performed with NEURON (*Carnevale and Hines, 2006*), version 7.1. In light of the evidence that both $\alpha_{1DS}$ and $\alpha_{1DL}$ could express in SNc neurons, we simulated $Ca_V1.3$ of two extreme settings with distinct inactivation kinetics: one is lack of CDI (ultrastrong CMI) and the other exhibits pronounced CDI representing $\alpha_{1DS}$ (without CMI). CMI effects including the endogenous CMI as in $\alpha_{1DL}$ could then be simulated by adjusting the relative weights of the above two states, resulting into varying (intermediate) levels of $S_{Ca}$ and $J_{Ca}$ but with constant $I_{300}$ throughout. Subsequently, we implemented this new $Ca_V1.3$ model and substituted the original L-type $Ca^{2+}$ current mechanism in a published Neuron model of SNc (*Chan et al., 2007*), where amendments were incorporated to appropriately reproduce oscillation and pacemaking (*Guzman et al., 2009*; *Putzier et al., 2009*). To simulate CCB effects, $I_{Ca}$ and its maximum conductance were decreased by ~30%.

## Data analysis and statistics

Data were analyzed in Matlab and Origin (Origin Software, San Clemente, CA). The standard error of the mean (S.E.M.) and Student's t-test (unpaired; two-tailed with criteria of significance: *p<0.05; **p<0.01; ***p<0.001) were calculated when applicable.

## Acknowledgements

We thank all X-Lab (Liu lab) members for discussions. We acknowledge the researchers who shared constructs and the cell line as indicated in the Methods section. This work is supported by Natural Science Foundation of China (NSFC) grants 81171382, 31370822 and 81371604; Beijing Natural Science Foundation (BNSF) grant 7142089; National Institutes of Health grant (RO1 GM107585); Tsinghua National Lab for Information Science and Technology (TNList) Cross-discipline Foundation; and open funds from Key Laboratory for Biomedical Engineering of Ministry of Education, Zhejiang University, China and from CAS Key Laboratory of Receptor Research, Shanghai, China. XDL also receives support from Tsinghua-Peking Center of Life Sciences.

## Additional information

### Funding

| Funder | Grant reference number | Author |
|---|---|---|
| National Natural Science Foundation of China | 81171382 | Xiaodong Liu |
| Natural Science Foundation of Beijing Municipality | 7142089 | Xiaodong Liu |
| National Natural Science Foundation of China | 81371604 | Xiaodong Liu |
| National Natural Science Foundation of China | 31370822 | Xiaodong Liu |
| National Institutes of Health | RO1 GM107585 | Henry M. Colecraft |

The funders had no role in study design, data collection and interpretation, or the decision to submit the work for publication.

### Author contributions

NL, Data curation, Formal analysis, Writing—original draft; YXY, Data curation, Formal Analysis, Experimental Design; LG, Data curation, Formal analysis, Conducted CaV1.3 and SNc modeling; ML, Data curation, Preliminary analysis; HMC, Provided materials and expertise for chemcial-inducible assay; XDL, Conceptualization, Supervision, Formal analysis, Writing—original draft, Writing—review and editing.

**Author ORCIDs**

Nan Liu, http://orcid.org/0000-0002-9606-4732

Xiaodong Liu, http://orcid.org/0000-0002-3171-9611

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
