## [Decision Letter]

Thank you for submitting your article "Cooperative and acute inhibition by multiple C-terminal motifs of L-type Ca^2+^ channels" for consideration by *eLife*. Your article has been favorably evaluated by Richard Aldrich (Senior Editor) and three reviewers, one of whom, Kenton J Swartz (Reviewer #1), is a member of our Board of Reviewing Editors.

The reviewers have discussed the reviews with one another and the Reviewing Editor has drafted this decision to help you prepare a revised submission.

Summary:

This manuscript reports interactions among C-terminal structural components of L-type Ca channels IQ, PCRD and DCRD, and the effects of such interactions on voltage gated activation (VGA) and CaM mediated Ca dependent inactivation (CDI). The interactions among IQ, PCRD and DCRD can alter channel function (known as CMI) only when any two of them are tethered together by fusion protein or rapamycin-mediated heterodimerization. The spatial closeness of any two of these structural domains may strengthen their interactions enough to compete off the IQ-CaM interaction, which is suggested by experiments with reduced apoCaM and live cell FRET tw0-hybrid assays. This mechanism directly explains why the interactions among IQ, PCRD and DCRD reduce CDI. Previous studies suggest that CDI may actually close the channel by reducing activation and results in this study show that at the end of CDI and CMI the currents levels are the same, the overall results suggest that CMI may alter channel function similarly as CDI. The mechanism revealed by this manuscript explains well the different phenotypes of various L-type Ca channels. By computational modeling, the manuscript suggests physiological importance of CMI, and rapamycin-mediated heterodimerization experiments provided a novel rationale for pharmacological intervention of Ca channel function. The study is significant and the results in general provide a strong argument for the proposed mechanism.

Essential revisions:

1) An overarching and major weakness of the manuscript is the lack of clarity in the written presentation. The text requires extensive editing to be acceptable for publication in *eLife*. The authors need to provide clear and logical rationale for each of the experiments performed and for what the results mean. The current presentation assumes that the reader is familiar with the authors previous work, as well as an extensive body of work on CDI. The authors need to undertake a major revision of the text to make the work accessible to a broad audience, and for the authors to provide more details on how each experiment was performed. The Abstract is full of hyperbole and needs to be rewritten to convey the main findings of the study and why they are important. We emphasize that we think this is an interesting study, but that the written presentation needs a lot of work. The comments below give more specific guidance.

2) The manuscript uses a lot of jargon and abbreviations, but does not give clear definition for all of them. The authors need to clearly explain, using correct and simple English sentences, in the beginning of Results the important molecular processes and parameters, such as CMI, CDI, S_Ca_, VGA, J_Ca_ and r_50_ etc., and use both text and figures to illustrate the definition, the experimental phenomena of these processes, and how they are measured.

3) Since CMI inhibits both voltage gated activation (VGA) and Ca dependent inactivation (CDI), it is too simplistic to measure CMI only based on CDI size (Figure 2, Figure 4). It would be better to measure both VGA and CDI at various voltages and the variation in VGA and CDI changes observed with different experimental conditions need better explanation and discussion.

4) The authors tend to make conclusions by simply referring to figures without convincing analyses or clear explanation. The authors should describe the most relevant feature of the result in the text and point out how the result leads to the conclusion. The authors should also explain clearly if the conclusion is novel or seemingly at odds with previously established mechanisms.

5) Structural domains from α_1F_ and α_1D_ such as PCRD-F and PCRD-D are intermittently used in different experiments without justification or explanation. In the text or figure legend, prior to the description of results, briefly explain why D or F form is used; do not label D or F if the two forms do not make any difference to the result.

6) Figure 2. CMI is supposed to inhibit both activation and CDI, but in Figure 2 the right two panels only show a reduction of CDI but not activation. The authors should explain and discuss this.

7) Do the authors suggest that in Figure 2, with IQ or IQ/PCRD, the channel can have CDI without CaM?

8) Figure 3. There is no experimental evidence that PCRD or DCRD were actually expressed separately.

9) There is a lack of clarity about the chimeric constructs used. The authors explain that they will "employ homologous DCTF from α1F with the strongest CMI among Ca_V_1 family to construct chimeric Ca_V_1.3 channels by fusing PCRDF-DCRDF onto IQ motif of α1DS". However, some of the experiments are conducted with PCRDD (Figure 1, Figure 2, Figure 4, Figure 6) without giving an explanation or rationale.

10) The quantification of the FRET experiments is missing: what is the equation fitting FRET strength (as in Figure 4, etc.)? Please report fitting parameters.

11) The rapamycin experiments are elegant and insightful. However, it is not clear why 300ms pulses were chosen, as it does not appear steady state for CMI (Figure 6 red current traces). This is also evident in Figure 1, where the current does not seem to level off at 300ms. The end stage of CMI and CDI may not be the same with longer pulses (true steady-state). This point seems crucial to the authors' conclusion that CDI and CMI functionally result into indistinguishable gating at the ending stage. Experiments with longer depolarizing pulses or measurements of CDI and CMI time constants may help to discuss this point.

---

## [Author Response]

*Essential revisions:*

*1) An overarching and major weakness of the manuscript is the lack of clarity in the written presentation. The text requires extensive editing to be acceptable for publication in eLife. The authors need to provide clear and logical rationale for each of the experiments performed and for what the results mean. The current presentation assumes that the reader is familiar with the authors previous work, as well as an extensive body of work on CDI. The authors need to undertake a major revision of the text to make the work accessible to a broad audience, and for the authors to provide more details on how each experiment was performed. The Abstract is full of hyperbole and needs to be rewritten to convey the main findings of the study and why they are important. We emphasize that we think this is an interesting study, but that the written presentation needs a lot of work. The comments below give more specific guidance.*

We have extensively revised the text according to the suggestions and comments.

*2) The manuscript uses a lot of jargon and abbreviations, but does not give clear definition for all of them. The authors need to clearly explain, using correct and simple English sentences, in the beginning of Results the important molecular processes and parameters, such as CMI, CDI, S_Ca_, VGA, J_Ca_ and r_50_ etc., and use both text and figures to illustrate the definition, the experimental phenomena of these processes, and how they are measured.*

We have revised the definitions by using text and figures, for the major processes of CDI, VGA and CMI, and key parameters of S_Ca_and J_Ca_. Also, we have provided the descriptions and illustrations for the amplitudes (I) and densities (J) of Ca^2+^ and/or Ba^2+^ currents measured at different time points, i.e., instantaneous peak, 50 ms, 300 ms and 1000 ms. Specifically,

A) Figure 1 and subsection “Provisional CMI concurrently attenuates both CDI and VGA”: descriptions on CDI and VGA;

B) The Introduction, subsection “Provisional CMI concurrently attenuates both CDI and VGA” and subsection “CMI is a unique type of inhibition but sharing similarities with CDI”: the introduction, basic characterization and eventual establishment for CMI;

C) Figure 1 and subsection “Provisional CMI concurrently attenuates both CDI and VGA”: I_Ca_and I_Ba_; S_Ca_and J_Ca_; I_peak_and J_peak_; I_50_and r_50_; D) Figure 6—figure supplement 1, Figure 6—figure supplement 2 and subsection “CMI is a unique type of inhibition but sharing similarities with CDI”: I_300_ and I_1000_.

*3) Since CMI inhibits both voltage gated activation (VGA) and Ca dependent inactivation (CDI), it is too simplistic to measure CMI only based on CDI size (Figure 2, Figure 4). It would be better to measure both VGA and CDI at various voltages and the variation in VGA and CDI changes observed with different experimental conditions need better explanation and discussion.*

We have revised all related figures to include both VGA and CDI (with the indices of J_Ca_and S_Ca_), which are Figure 2, Figure 2—figure supplement 1, Figure 3 and Figure 3—figure supplement 2, and Figure 4. We agree with the reviewer about measuring both VGA and CDI. Technically, CDI and its index SCa are more favorable because for Ca_V_1.3 channels CDI is independent of global Ca^2+^ so insensitive to buffer capacities, variations in expression levels, or the general health of the cells, thus more robust than VGA (J_Ca_). In general, VGA and J_Ca_are less stable even when experimental conditions are well controlled. Basically, any conclusion on J_Ca_needs to be carefully drawn from pronounced changes. In fact, this was one of the motivations underlying chemical-inducible CMI, for us to confidently count on each individual cell to quantify VGA and other properties. In some cases, the mild changes in J_Ca_may lead to uncertainties which need to be confirmed by analyzing concurrent CDI changes. We have discussed these aspects with relevant data (subsection “Differential roles of DCRD and PCRD unveiled by low [apoCaM]” and Figure 2—figure supplement 1).

*4) The authors tend to make conclusions by simply referring to figures without convincing analyses or clear explanation. The authors should describe the most relevant feature of the result in the text and point out how the result leads to the conclusion. The authors should also explain clearly if the conclusion is novel or seemingly at odds with previously established mechanisms.*

We have paid attention to these aspects in the revision, by providing more details of the experiments and data in the main text in addition to the figure captions, especially for Figure 4–Figure 6, which represent the major conclusions in this work (subsections “Cooperative scheme of CMI consists of three major combinations”, “Acute CMI based on rapamycin-inducible heterodimerization and cooperation” and “CMI is a unique type of inhibition but sharing similarities with CDI”). Also, we thank the reviewer for the suggestion about scrutinizing the novelty of particular results. It has been well taken to emphasize on the key findings of this work and we have revised the relevant text accordingly, to list a few:

A) Subsection “Differential roles of DCRD and PCRD unveiled by low [apoCaM]” and Figure 2, surprising observations from α_1DS_-G_X_-DCRD that CDI and VGA were attenuated at low [apoCaM];

B) Subsection “Cooperative scheme of CMI consists of three major combinations” and Figure 4, novel roles of PCRD unveiled by effective CMI (combination III);

C) Subsection “CMI is a unique type of inhibition but sharing similarities with CDI” and Figure 6, findings about stable end-stage CDI (I_300_) from both versions of inducible CMI, distinct from known modalities such as run-down and small-molecule blockage;

D) Subsection “CMI is a unique type of inhibition but sharing similarities with CDI”and Figure 6, new discovery that CMI attenuation on CDI is achieved by inhibiting the peak amplitude of Ca^2+^ current while maintaining its late-phase levels, which also reconciles the concurrent VGA/CDI attenuations apparently leading to contradictory effects on Ca^2+^ influx.

*5) Structural domains from α_1F_ and α_1D_ such as PCRD-F and PCRD-D are intermittently used in different experiments without justification or explanation. In the text or figure legend, prior to the description of results, briefly explain why D or F form is used; do not label D or F if the two forms do not make any difference to the result.*

We agree with the reviewer that it is unnecessary in the context of this work to distinguish between PCRD_D_and PCRD_F_(subsection “Provisional CMI concurrently attenuates both CDI and VGA”). We have updated the sequence alignment for homology among different Ca_V_1 isoforms including α_1D_ and α_1F_ (Figure 1—figure supplement 1). And the functional roles of the two motifs in CMI are essentially very close, because PCRD_D_-DCRD and PCRD_F_-DCRD peptides attenuated α_1DS_ similarly (Figure 4—figure supplement 1 and subsection “Cooperative scheme of CMI consists of three major combinations”). Following the suggestion, we have also simplified the nomenclature for DCRD and IQ_V_, since in this work either only one isoform was used (DCRD_F_); or even different isoforms were used but functionally similar (PCRD_D_ vs. PCDR_F_; IQ_V_ from Ca_V_1.3 vs. Ca_V_1.2). Meanwhile, details on various constructs including their original templates have been provided in the section of “Materials and methods” (“Molecular biology”).

*6) Figure 2. CMI is supposed to inhibit both activation and CDI, but in Figure 2 the right two panels only show a reduction of CDI but not activation. The authors should explain and discuss this.*

We thank the reviewer for the comment with specifics. As the reviewer pointed out, it is true that both CDI and activation (VGA) were inhibited in the mentioned cases. Due to the reasons as stated in item 3, we prefer to take advantages of CDI (S_Ca_) as the major index to examine CMI effects and thus we did not show VGA data. In response to this comment and also for further assurance, we have provided full profiles of VGA (Figure 2—figure supplement 1), with J_Ca_values summarized in Figure 2. These results have also been described accordingly in the text (subsection “Differential roles of DCRD and PCRD unveiled by low [apoCaM]”).

*7) Do the authors suggest that in Figure 2, with IQ or IQ/PCRD, the channel can have CDI without CaM?*

BSCaM_IQ_, a specific apoCaM (Ca^2+^-free CaM) chelator, was to substantially reduce free apoCaM levels. However, in practice, BSCaM_IQ_ cannot eliminate all the free apoCaM in cells. Since CaM is generally needed, supposedly no cells with complete elimination of apoCaM would be available for experimental assessment. We have modified the illustration (Figure 2) accordingly. Under the residue apoCaM in realistic cells, without DCRD competition, α_1DS_ or α_1DS_-PCRD is still able to produce pronounced CDI similarly as in control conditions, owing to the fact that under these conditions apoCaM binds the channels (IQ_V_ or IQ_V_-PCRD) reasonably well (Liu et al. 2010). Although such low apoCaM would not affect α_1DS_ or α_1DS_-PCRD, it does reduce the CDI of some variants besides α_1D_-G_X_-DCRD, such as the native α_1DL_ variants containing full-length DCTs. The established principles for CaM-mediated CDI are still valid and applicable in all these cases, regardless of actual CaM levels. We have revised the text to clarify this notion (subsection “Differential roles of DCRD and PCRD unveiled by low [apoCaM]”).

*8) Figure 3. There is no experimental evidence that PCRD or DCRD were actually expressed separately.*

For the peptide experiments in Figure 3, both PCRD and DCRD were tagged with fluorescent proteins. And in the case of PCRD/DCRD coexpression, PCRD and DCRD were fused with YFP vs. CFP to make sure both were present in the same cell (Figure 3—figure supplement 1 and subsection “Cooperative scheme of CMI consists of three major combinations”). We have amended the construct names with explicit information of the tags in Figure 3and also other peptide experiments (mainly in Figure 4and Figure 5). In addition, subsequent inducible-CMI experiments (e.g., Figure 5) demonstrate that the ultrastrong CDI as in control was substantially attenuated upon induced PCRD-DCRD dimerization, which provides another proof that both peptides can be separately expressed very well in the same cell without affecting CDI of α_1DS_.

We assume this comment is also concerned with the potential binding/assembly between the two individual peptides of PCRD and DCRD, which should be unlikely according to this work and previous reports. For PCRD and DCRD peptides, no binding was detected with the FRET two-hybrid assay (Liu et al. 2010). Nevertheless, this would not completely exclude all potential interactions between PCRD and DCRD. For instance, electrostatic interactions might exist between PCRD and DCRD in the holo-channel of Ca_V_1.2 (Hulme et al. 2005), provided that the major binding has been established in the first place, e.g., between IQ_V_-PCRD and DCRD. The important consensus here is that even if the putative PCRD/DCRD interactions exist, they should be weaker than the binding between IQ_V_ and DCRD (Figure 4); and the latter binding is further enhanced by the presence of PCRD in the close vicinity to IQ_V_ or DCRD, which might be partly contributed by PCRD/DCRD interaction. We have revised the text to incorporate some of the above facts and analyses (subsection “Cooperative scheme of CMI consists of three major combinations”).

*9) There is a lack of clarity about the chimeric constructs used. The authors explain that they will "employ homologous DCTF from α1F with the strongest CMI among Ca_V_1 family to construct chimeric Ca_V_1.3 channels by fusing PCRDF-DCRDF onto IQ motif of α1DS". However, some of the experiments are conducted with PCRDD (Figure 1, Figure 2, Figure 4, Figure 6) without giving an explanation or rationale.*

Please see item 5.

*10) The quantification of the FRET experiments is missing: what is the equation fitting FRET strength (as in Figure 4, etc.)? Please report fitting parameters.*

The fitting procedures were based on established protocols and algorithms (Erickson et al. 2001; Liuet al. 2010; Butz et al. 2016). In the “Materials and methods” section (“FRET optical imaging”) we have added a brief description of the FRET two-hybrid binding assay (3-cube FRET) used in this work. We appreciate the reviewer for this suggestion reminding us to provide the values for the key binding parameters (effective dissociation equilibrium constant K_d_ indicating affinity and also the maximum FRET ratio FR_max_ intrinsic to the bound pair, which we obtained from the iterative curve fitting procedures with a customized Matlab program. Accordingly, for all the pairs of FRET binding, K_d_ and FR_max_ values have been included for Figure 4 in the caption and the main text (subsection “Cooperative scheme of CMI consists of three major combinations”), and in Figure 4—figure supplement 2, and Figure 6—figure supplement 3.

*11) The rapamycin experiments are elegant and insightful. However, it is not clear why 300ms pulses were chosen, as it does not appear steady state for CMI (Figure 6 red current traces). This is also evident in Figure 1, where the current does not seem to level off at 300ms. The end stage of CMI and CDI may not be the same with longer pulses (true steady-state). This point seems crucial to the authors' conclusion that CDI and CMI functionally result into indistinguishable gating at the ending stage. Experiments with longer depolarizing pulses or measurements of CDI and CMI time constants may help to discuss this point.*

We thank the reviewer for bringing up this important matter. We have conducted the experiments with longer depolarization pulses as suggested for both constitutive and rapamycin-inducible CMI (Figure 6—figure supplement 2). Briefly, for Ca_V_1.3 channels, 300 ms is already very close to the end stage or the steady state of CDI; and with both 300 ms and 1000 ms pulses, the “end-stage” CMI (or full- strength CMI) is very close to the end-stage CDI (or steady-state CDI).

The decreasing trend or tendency observed from the Ca^2+^ current at 300 ms is largely due to VDI (voltage-dependent inactivation), as confirmed by the comparison between current traces in Ba^2+^ and Ca^2+^. VDI varies slightly from cell to cell, but generally much slower than CDI so often observable at 300 ms or even later phase during the depolarization step. Therefore, “unsteady” I_Ca_ is not necessarily a sign of incomplete CDI.

Meanwhile, we agree with the reviewer that longer pulse such as 1000 ms would be informative and potentially more confirmative. Consistent with J_300_-based results, J_1000_ (Ca^2+^ current density measured at 1000 ms) profiles from α_1DS_ control vs. α_1DS_-PCRD-DCRD were nearly identical, both suggesting that end-stage CDI remained unaltered even with potent CMI attenuation (Figure 6—figure supplement 2). For rapamycin-inducible CMI, both I_1000_ and I_300_ remained stable throughout the experiments (Figure 6—figure supplement 2), as compared to decaying I_peak_ and I_100_ (measured at 0 and 100 ms respectively) (Figure 6—figure supplement 2). Taken together, the performance of I_300_ is comparable to I_1000_, so both are qualified indices of end-stage CDI (for channels with or without CMI) (Figure 6—figure supplement 2).

We would like to clarify that “end-stage” of inducible CMI refers to its steady state in the time-dependent profile, and its actual level of attenuation is dependent on the potency of particular CMI competition. For constitutive CMI, “end-stage” CMI refers to the full-strength CMI or complete competition, which was closely approached but not yet fully reached in most experiments (but see very strong CMI effects in Figure 1 and Figure 4). In contrast, end-stage CDI in this work always refers to the time domain, reaching its steady state at ~300 ms or later. As suggested by the whole-cell I_Ca_ from ensemble channels, we have concluded that the particular channel inhibited by CMI (with CDI abolished) should be functionally equivalent to the channel (e.g., with no CMI) into its end-stage CDI. Provided that VDI is negligible, I_Ca_ attenuated by full-strength CMI (evaluated at 0 ms, I_peak_) would be equal to I_Ca_ at the end stage of CDI (≥300 ms, e.g., I_300_), as our data of rapamycin-inducible CMI (version 2) strongly suggest (Figure 6 and Figure 6—figure supplement 2).

Some of these discussions have been included in the revision (Figure 6, subsection “CMI is a unique type of inhibition but sharing similarities with CDI”).